# Core functional nodes and sex-specific pathways in human ischaemic and dilated cardiomyopathy

Mengbo Li [1,2,3,10], Benjamin L. Parker[3,4,10], Evangeline Pearson[2,3], Benjamin Hunter[2,5], Jacob Cao[2,5], Yen Chin Koay[2,3,6], Oneka Guneratne[2,5], David E. James[3,7,8], Jean Yang [1,3], Sean Lal [2,5,8,9 ✉] & John F. O'Sullivan [2,3,6,8,9 ✉]

Poor access to human left ventricular myocardium is a significant limitation in the study of heart failure (HF). Here, we utilise a carefully procured large human heart biobank of cryo-preserved left ventricular myocardium to obtain direct molecular insights into ischaemic cardiomyopathy (ICM) and dilated cardiomyopathy (DCM), the most common causes of HF worldwide. We perform unbiased, deep proteomic and metabolomic analyses of 51 left ventricular (LV) samples from 44 cryopreserved human ICM and DCM hearts, compared to age-, gender-, and BMI-matched, histopathologically normal, donor controls. We report a dramatic reduction in serum amyloid A1 protein in ICM hearts, perturbed thyroid hormone signalling pathways and significant reductions in oxidoreductase co-factor riboflavin-5-monophosphate and glycolytic intermediate fructose-6-phosphate in both; unveil gender-specific changes in HF, including nitric oxide-related arginine metabolism, mitochondrial substrates, and X chromosome-linked protein and metabolite changes; and provide an interactive online application as a publicly-available resource.

[1] School of Mathematics and Statistics, Faculty of Science, The University of Sydney, Sydney, NSW, Australia. [2] Precision Cardiovascular Laboratory, The University of Sydney, Sydney, NSW, Australia. [3] Charles Perkins Centre, School of Life and Environmental Sciences, The University of Sydney, Sydney, NSW, Australia. [4] Department of Physiology, School of Biomedical Sciences, The University of Melbourne, Melbourne, VIC, Australia. [5] Discipline of Anatomy and Histology, School of Medical Sciences, The University of Sydney, Sydney, NSW, Australia. [6] Heart Research Institute, The University of Sydney, Sydney, NSW, Australia. [7] School of Life and Environmental Sciences, Faculty of Science, The University of Sydney, Sydney, NSW, Australia. [8] Central Clinical School, Sydney Medical School, Faculty of Medicine, The University of Sydney, Sydney, NSW, Australia. [9] Department of Cardiology, Royal Prince Alfred Hospital, Camperdown, NSW, Australia. [10] These authors contributed equally: Mengbo Li, Benjamin L. Parker. ✉email: sean.lal@sydney.edu.au; john.osullivan@hri.org.au

Heart failure (HF) is the result of complex interaction between genetic predisposition and environmental triggers. The paucity of human left ventricular myocardial tissue for research is a major limitation to directly studying heart failure pathogenesis. Our extensive and heart bank has been carefully procured over the last 30 years, and novel findings in our have been replicated by numerous and unrelated groups, and cross-validated with samples derived elsewhere and in model systems[1–8].

The two leading causes of cardiac transplantation are ischaemic (ICM) and dilated cardiomyopathy (DCM). ICM has a clear causal precipitant (myocardial ischaemia), and post-ischaemic remodelling is well-described, yet widely variable[9]. The core feature of DCM is LV dilatation and systolic dysfunction in the absence of coronary artery disease and abnormal loading conditions, and underlying aetiologies include genetic, inflammatory, metabolic, infiltrative, and autoimmune[10]. In both ICM and DCM, disease progression is highly variable and unpredictable. Molecular drivers underpinning DCM, in particular, have remained largely elusive, and the reason differing aetiological origins result in a common DCM phenotype is poorly understood[11].

Studies examining the functional genomic signature of DCM and ICM explanted hearts have generally concentrated on "upstream" domains[12], incorporating whole-genome sequencing, transcriptomic profiling, and sometimes protein expression. This work has provided novel insights into molecular changes in human heart failure, implicating extracellular matrix remodelling, inflammatory signalling, oxidative stress, mitochondrial dysfunction, and branched chain amino acid metabolism as signature pathogenic changes[13–15]. The proteomic studies usually used small numbers, infrequently more than one aetiology, and without matching across age, gender, and BMI[13,14,16–20]. Further, a more extensive understanding as to whether upstream perturbations are propagated downstream via translation to the protein level, and further via enzymatic processing to the metabolite level, are required. Such analyses require large sample sizes, in addition to matched control (non-diseased) groups to adequately address the major confounders of age, gender, and body-mass index (BMI). Therefore, we determined the perturbations in ICM and DCM at the protein and metabolite pathway level using an unbiased and comprehensive screen of a large number of heart failure samples matched to histopathologically-normal donor controls for age, gender, and BMI—a total of 51 left ventricular myocardial samples from 44 human hearts (Table 1; Fig. 1a–c). Quantification of proteins and metabolites, both downstream of genetic variation, captures the contribution of genes, environment, and their interaction, and serves as a rich source of "translatable" diagnostic and therapeutic targets. Furthermore, multi-omic integration reveals interplay between different layers of a biological system such as metabolites with enzymes/transporters[21], and can discover new associations between these biological layers. We provide all proteomic and metabolomic results via an interactive online repository (https://mengboli.shinyapps.io/heartomics/) as a publicly available resource, thereby enabling researchers without access to human cardiac tissue.

## Results

**Patient characteristics.** The ages and BMIs of the donor, ICM, and DCM groups were similar (Table 1) (Fig. 1b). Gender distribution was evenly matched in donors, whereas both ICM and DCM groups had increased percentages of males in keeping with the epidemiology of these diseases (Fig. 1c). All HF patients were NYHA class III/IV with severely impaired left ventricular systolic function (reduced ejection fraction).

**Table 1 Summary of patient characteristics.**

| Characteristic | Donor (n = 15) | ICM (n = 15) | DCM (n = 14) |
|---|---|---|---|
| Male (%) | 46.7 | 66.7 | 66.7 |
| Age (mean) (years) | 53.2 | 54.2 | 55.3 |
| BMI (mean) (kg/m²) | 25.8 | 26.6 | 22.5 |
| Minimum EF (mean) (%) | >55 | 27.57 | 16.13 |
| NYHA class | N/A | III/IV | III/IV |
| LVEDD (mean) (mm) | <55 | 67.50 | 75.17 |
| TPG (mean) (mmHg) | N/A | 11.1 | 9.8 |
| Hypercholerolemia (mean) (%) | 20.0 | 20.0 | 21.4 |
| Kidney disease (mean) (%) | 0.0 | 6.7 | 21.4 |
| Family history of HF (mean) (%) | 6.7 | 13.3 | 35.7 |
| Diabetes (mean) (%) | 0.0 | 33.3 | 13.3 |
| Hypertension (mean) (%) | 6.7 | 26.7 | 21.4 |
| HF Medications | | | |
| Aldosterone antagonist (%) | 0.0 | 20.0 | 42.9 |
| Beta blocker (%) | 0.0 | 26.7 | 50.0 |
| ACE inhibitor (%) | 0.0 | 33.3 | 28.6 |

Complete clinical data was not available for every patient.
*EF* ejection fraction, *NYHA* New York Heart Association Functional Classification, *LVEDD* left ventricular end-diastolic dimension, *TPG* trans-pulmonary pressure gradient, *HF* heart failure, *ACE* angiotensin converting enzyme.

**Proteomics of ICM and DCM**. Quantitative proteomic analysis of heart tissue was performed by analysing tryptic peptides by ultra-high performance liquid chromatography with data-independent acquisition mass spectrometry (UHPLC-DIA-MS) coupled to identification with a spectral library generated in-house consisting of 6,182 unique proteins. UHPLC-DIA-MS resulted in the identification of 3,264 proteins with 2,614 quantified in >33 samples (75% data completion).

We initially assessed clustering of the groups via principal component analysis (PCA) (Fig. 2a). In total, 185 and 377 proteins were differentially expressed (DE) between ICM vs Donor and DCM vs Donor hearts, respectively (generalised linear model adjusted for gender with Benjamini–Hochberg adjusted $P$ value <0.05, Fig. 2b–d and Supplementary Data 1–3. Details on the fitted model are provided in the statistics section of Methods). The core set of proteins consisted of 153 proteins regulated in both ICM and DCM while 32 and 224 were unique to ICM or DCM, respectively (Fig. 2d).

The fold changes of proteins in ICM vs Donor, DCM vs Donor, and both vs donor are highlighted in Fig. 2e. Serum amyloid A1 (SAA1) was the most downregulated protein in both ICM and DCM (see below). Osteoglycin (OGN), important in the extracellular matrix and contributing to collagen deposition and fibrillogenesis in the heart[22,23], and pro-fibrotic proline-rich protein 12 (PRR12)[24], were significantly upregulated in both ICM and DCM. Periostin (POST), associated with mesenchymal differentiation in the heart[25], was also significantly upregulated in both ICM and DCM. A heatmap (Fig. 2f) summarising the differential proteomic data is consistent with the PCA data, with hierarchical clustering revealing good separation of donor samples from ICM or DCM, and co-regulation across a wide range of proteins, with dramatic elevation in certain proteins, e.g. collagen.

The top DE protein in ICM myocardium compared to donor was serum amyloid A1 (SAA1), which was reduced 10-fold ($P = 2.8 \times 10^{-7}$). SAA1 was also significantly decreased in DCM (FC = −5.3, $P = 4.7 \times 10^{-5}$) (Fig. 2e). There were significant changes in expression of a number of extracellular matrix (ECM) proteins (Table 2 and Supplementary Data 1): fibulin 5 (FBLN5) (FC = 2.7, $P = 1.16 \times 10^{-5}$), EGF-containing fibulin-like extracellular matrix protein 1 (EFEMP1) (FC = 3.4, $P = 1.1 \times 10^{-4}$) and microfibrillar-associated protein 4 (MFAP4), a component of elastin fibres (FC = 3.14, $P = 1.7 \times 10^{-4}$), which were all also

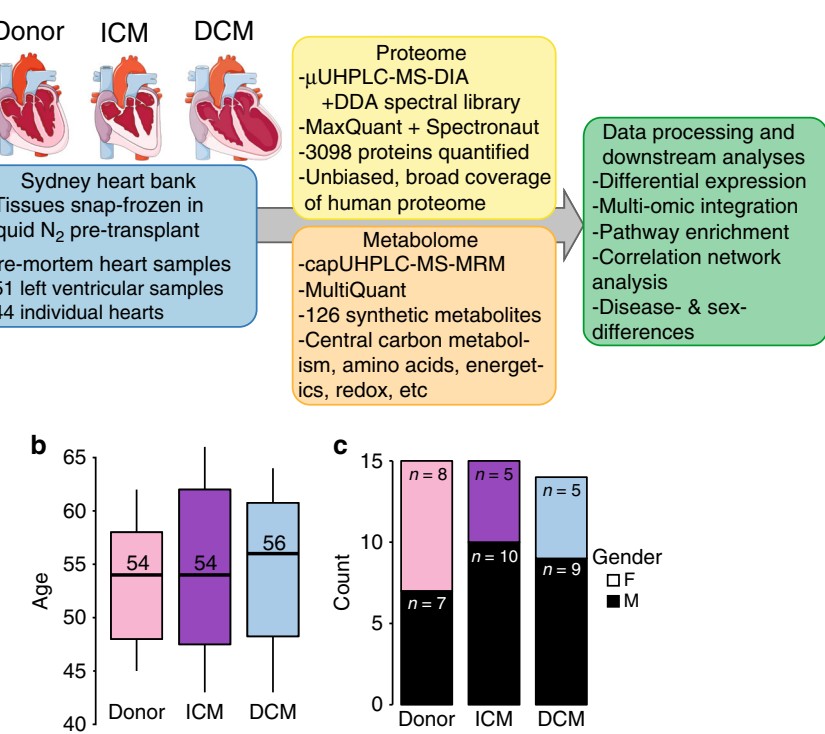

**Fig. 1 Schematic summary. a** Schematic outlining pathology, sample acquisition, omic platforms applied, and analysis. **b** age distribution of all measured patients and donors. The size of each condition group is as indicated in (**c**); the boxplots indicate the median (the middle line) and third and first quartiles (the box); the whiskers show the 1.5 × IQR (interquartile range) above and below the box. and (**c**) gender counts of donor heart, ischemic heart disease, and dilated cardiomyopathy groups. ICM ischemic cardiomyopathy, DCM dilated cardiomyopathy. Source data are provided as a Source Data file.

significantly differential in DCM vs Donor hearts, indicative of common remodelling pathways. Consistent with fibrotic remodelling post myocardial ischemia, numerous collagen proteins were significantly increased in the myocardium of ICM patients. Many collagen protein subunits were also significantly increased in DCM myocardium. Other extracellular proteins regulated in ICM (and DCM) were osteoglycin (FC = 3.99, $P = 5.3 \times 10^{-4}$) and plexin domain containing protein 2 (FC = 1.7, $P = 1.3 \times 10^{-3}$) (Supplementary Data 1 and 2).

Transferrin, an iron transport protein, was significantly increased in ICM myocardium (FC = 2.23, $P = 4.5 \times 10^{-6}$), suggesting increased iron deposition in ICM, and was also increased in DCM. Von Willebrand Factor, involved in haemostasis and platelet adhesion, was also significantly increased in ICM myocardium (FC = 2.8, $P = 4.5 \times 10^{-6}$), as well as DCM. Components of the blood coagulation cascade were amongst the most significant regulated proteins in ICM, but not DCM, as kallistatin (SERPINA4) (FC = 2.6, $P = 1.6 \times 10^{-4}$), heparin cofactor II (SERPIND1) (FC = 2.3, $P = 3.6 \times 10^{-4}$) and kininogen 1 (FC = 1.8, $P = 2.4 \times 10^{-3}$) were all differential when comparing ICM and donor myocardium.

In total, 377 proteins were significantly different between DCM and Donor myocardium (Fig. 2d and Supplemental Table 2), the top 10 illustrated in Table 3.

The top DE protein in DCM vs donor was the ECM protein fibulin 5 (FC = 3.6, $P = 1.99 \times 10^{-7}$). Other highly regulated ECM proteins were microfibril associated protein 4 (MFAP4) (FC = 4.7, $P = 9.8 \times 10^{-7}$), EGF-containing fibulin-like extracellular matrix protein 1 (EFEMP1) (FC = 4.2, $P = 5.6 \times 10^{-6}$), plexin domain containing protein 2 (PLXDC2) (FC = 2.1, $P = 5.8 \times 10^{-6}$), heparan sulfate proteoglycan 2 (HSPG2) (FC = 1.5, $P = 2.6 \times 10^{-5}$), fibulin 1 (FC = 2.2, $P = 3.1 \times 10^{-5}$), prolargin (PRELP) (FC = 4.5, $P = 9.1 \times 10^{-7}$), lumican (LUM) (FC = 3.1,

$P = 2.3 \times 10^{-4}$), versican (VCAN, a proteoglycan) (FC = 3.7, $P = 3.3 \times 10^{-4}$), extracellular matrix protein 1 (ECM1) (FC = 2.1, $P = 3.3 \times 10^{-4}$), and osteoglycin (OGN) (FC = 4.2, $P = 1.8 \times 10^{-4}$).

As can be seen in the Top 10 DE proteins in DCM (Table 3) and in Supplementary Data 2, increases in collagen proteins in DCM hearts were common. Mitogen-activated protein kinase-1 (MAP2K1), which acts as an essential component of the MAP kinase signal transduction pathway, was significantly decreased in DCM hearts (FC = −1.4, $P = 3.1 \times 10^{-5}$) (Table 3 and Supplementary Data 2). As in ICM, transferrin (FC = 2.3, $P = 2.0 \times 10^{-6}$), the iron transport protein, was significantly increased in DCM, suggesting iron deposition is also a relevant pathogenic process in DCM. Also similar to ICM, von Willebrand Factor was significantly increased in DCM (FC = 2.9, $P = 2.4 \times 10^{-6}$), perhaps indicating increased platelet adhesion in DCM.

**Metabolomics of ICM and DCM.** Targeted metabolomics analyses were performed using ultra-high performance liquid chromatography systems with multiple reaction monitoring mass spectrometry (UHPLC-MRM-MS) coupled to identification with an internal library of authentic chemical standards. As for the proteomic data, we initially assessed clustering of the metabolomic data using PCA (Fig. 3a). The PCA plot (Fig. 3a) demonstrated, at the global metabolomic level, distinct separation of the donor hearts from ICM and DCM hearts. We next performed a multi-group differential expression analysis, adjusted for gender, to identify metabolites regulated between ICM vs donor, DCM vs donor and also to reveal a core set of regulated metabolites common to both cardiomyopathies. In total, 28 metabolites were DE between ICM vs Donor or DCM vs Donor hearts (Benjamini–Hochberg adjusted $P$ value <0.05, Fig. 3b-d and Supplementary Data 4–6). The core set of metabolites consisted

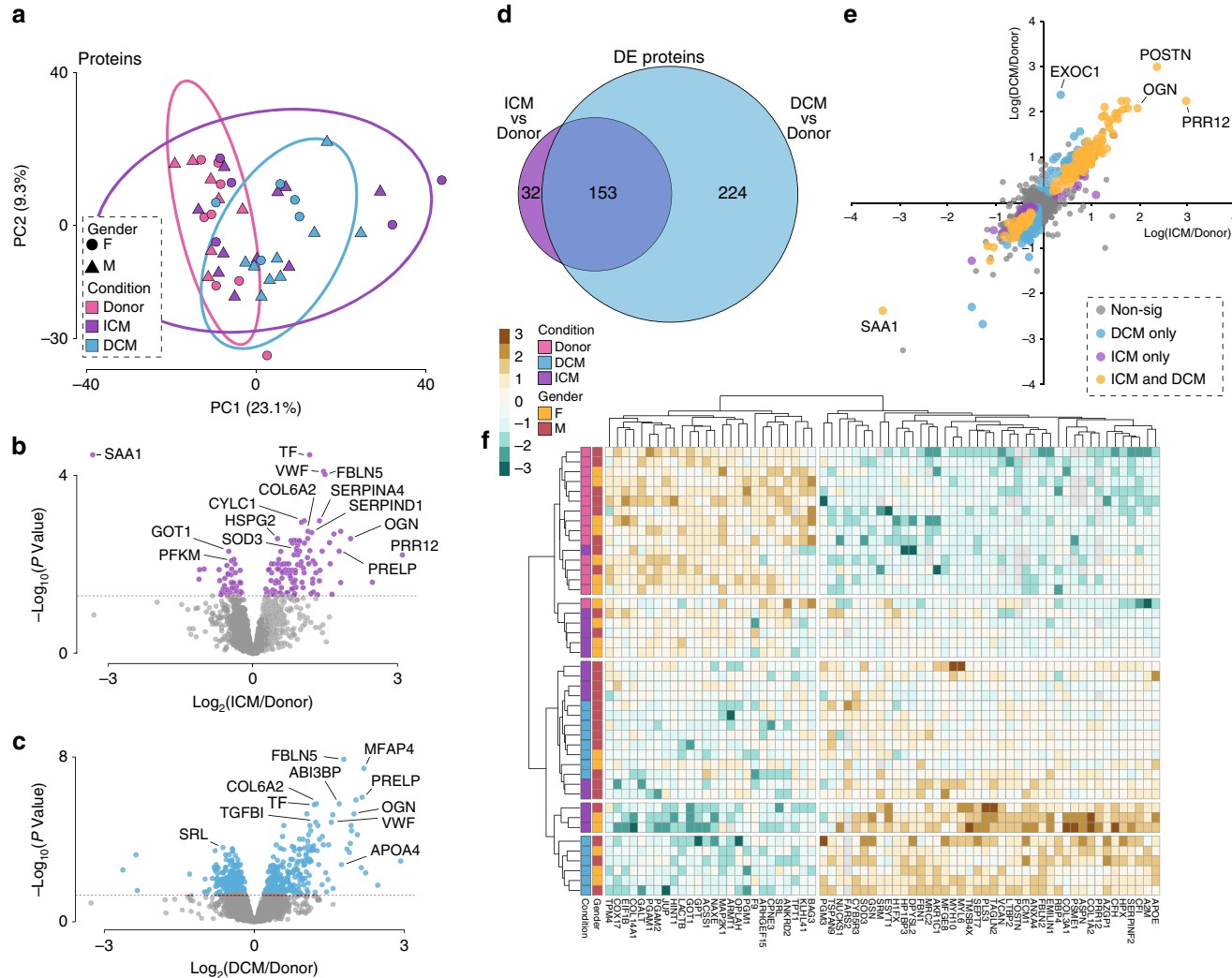

**Fig. 2 Differential analysis in protein expression levels between ICM or DCM and donors. a** PCA of proteomic data in ICM and DCM, including all three conditions and both genders. **b** Volcano plot of ICM vs Donor proteins. Estimates were derived using the linear regression model adjusted for gender, where $n_{ICM} = 15$ and $n_{donor} = 15$. **c** Volcano plot of DCM v Donor proteins. Estimates were derived using the linear regression model adjusted for gender, where $n_{DCM} = 14$ and $n_{donor} = 15$. **d** Venn diagram summarising the differential and overlapping proteins. **e** Plot of fold changes of proteins in ICM, DCM, and both. **f** Multigroup heatmap with dendrogram of DE protein levels incorporating donor, ICM, and DCM hearts and both genders. DE differential expression. Source data are provided as a Source Data file.

**Table 2 Top 10 DE Proteins in ICM vs Donor left ventricular myocardium.**

| Protein | Log2 FC | FC | t | P value | Adjusted P value |
|---|---|---|---|---|---|
| SAA1 | −3.32 | −10 | −8.52 | $1.08 \times 10^{-10}$ | $2.84 \times 10^{-07}$ |
| TF | 1.16 | 2.23 | 7.37 | $3.84 \times 10^{-09}$ | $4.53 \times 10^{-06}$ |
| VWF | 1.47 | 2.77 | 7.28 | $5.20 \times 10^{-09}$ | $4.53 \times 10^{-06}$ |
| FBLN5 | 1.45 | 2.74 | 6.91 | $1.77 \times 10^{-08}$ | $1.16 \times 10^{-05}$ |
| EFEMP1 | 1.78 | 3.43 | 6.26 | $2.06 \times 10^{-07}$ | $1.08 \times 10^{-04}$ |
| SERPINA4 | 1.35 | 2.55 | 6.12 | $3.59 \times 10^{-07}$ | $1.56 \times 10^{-04}$ |
| MFAP4 | 1.65 | 3.14 | 5.99 | $4.47 \times 10^{-07}$ | $1.67 \times 10^{-04}$ |
| COL6A2 | 1.14 | 2.20 | 5.80 | $7.21 \times 10^{-07}$ | $2.36 \times 10^{-04}$ |
| CYLC1 | 1.01 | 2.01 | 5.75 | $8.43 \times 10^{-07}$ | $2.45 \times 10^{-04}$ |
| SERPIND1 | 1.19 | 2.28 | 5.60 | $1.41 \times 10^{-06}$ | $3.57 \times 10^{-04}$ |

Estimates were derived using the linear regression model adjusted for gender, where $n_{ICM} = 15$ and $n_{donor} = 15$; p values were adjusted for multiple comparisons by the Benjamini–Hochberg method. (See Supplementary Data 1).
FC fold change, t t-statistic.

**Table 3 Top 10 DE Proteins in DCM vs Donor myocardium.**

| Protein | Log2 FC | FC | t | P value | Adjusted P value |
|---|---|---|---|---|---|
| FBLN5 | 1.83 | 3.55 | 8.57 | $7.60 \times 10^{-11}$ | $1.99 \times 10^{-07}$ |
| MFAP4 | 2.22 | 4.67 | 7.97 | $7.46 \times 10^{-10}$ | $9.75 \times 10^{-07}$ |
| TF | 1.20 | 2.30 | 7.52 | $2.30 \times 10^{-09}$ | $2.01 \times 10^{-06}$ |
| VWF | 1.51 | 2.86 | 7.38 | $3.64 \times 10^{-09}$ | $2.38 \times 10^{-06}$ |
| EFEMP1 | 2.06 | 4.18 | 7.18 | $1.08 \times 10^{-08}$ | $5.62 \times 10^{-06}$ |
| PLXDC2 | 1.05 | 2.08 | 6.73 | $3.23 \times 10^{-08}$ | $1.41 \times 10^{-05}$ |
| HSPG2 | 0.59 | 1.50 | 6.50 | $6.92 \times 10^{-08}$ | $2.58 \times 10^{-05}$ |
| FBLN1 | 1.11 | 2.16 | 6.41 | $9.43 \times 10^{-08}$ | $3.08 \times 10^{-05}$ |
| COL6A2 | 1.26 | 2.40 | 6.36 | $1.10 \times 10^{-07}$ | $3.11 \times 10^{-05}$ |
| MAP2K1 | −0.44 | −1.36 | −6.34 | $1.19 \times 10^{-07}$ | $3.11 \times 10^{-05}$ |

Estimates were derived using the linear regression model adjusted for gender, where $n_{DCM} = 14$ and $n_{donor} = 15$; p values were adjusted for multiple comparisons by the Benjamini–Hochberg method. (See Supplementary Data 2).
FC fold change, t t-statistic.

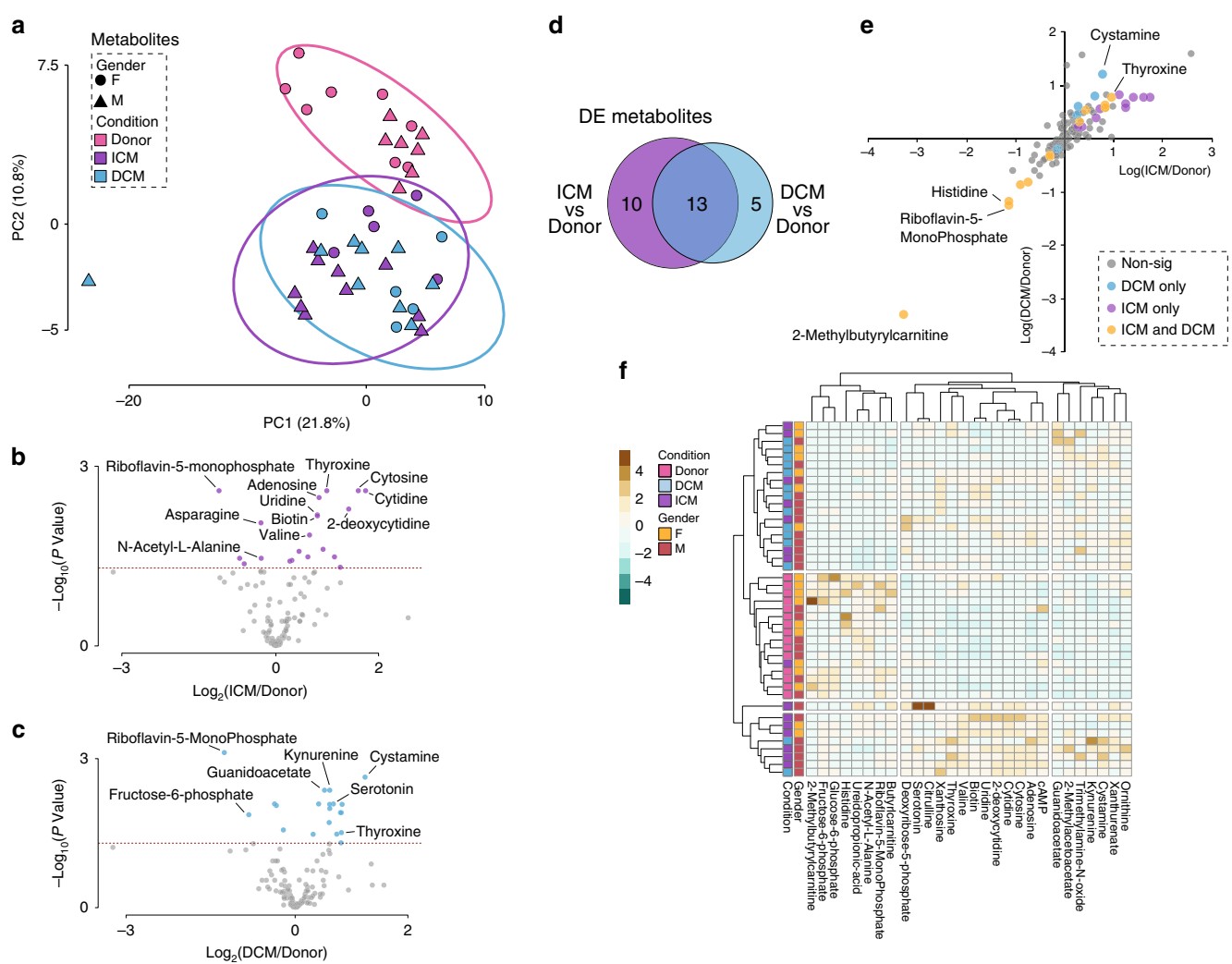

**Fig. 3 Differential analysis in metabolite abundance levels between ICM or DCM and donors. a** PCA of metabolomic data in ICM and DCM, including all three conditions and both genders. **b** Volcano plot of ICM vs Donor metabolites. Estimates were derived using the linear regression model adjusted for gender, where $n_{ICM} = 15$ and $n_{donor} = 15$. **c** Volcano plot of DCM v Donor metabolites. Estimates were derived using the linear regression model adjusted for gender, where $n_{DCM} = 14$ and $n_{donor} = 15$. **d** Venn Diagram summarising the differential and overlapping metabolites. **e** Plot of fold changes of metabolites in ICM, DCM, and both. **f** Multigroup heatmap with dendrogram of DE metabolite levels incorporating donor, ICM, and DCM hearts and both genders. Source data are provided as a Source Data file.

of 13 metabolites regulated in both ICM and DCM while 10 were unique to ICM and 5 unique to DCM (Fig. 3d). The most significant DE metabolite in both ICM (FC = 0.46, $P = 1.3 \times 10^{-4}$) and DCM (FC = 0.4, $P = 1.2 \times 10^{-5}$) hearts was the oxidoreductase co-factor riboflavin-5- monophosphate, also known as flavin mononucleotide (Tables 4, 5), suggesting oxidative stress is a prominent feature of both pathologies. Another mitochondrial substrate, 2-methylbutyrylcarnitine, had the greatest fold-change decrease in both ICM and DCM (Fig. 3e), indicative of mitochondrial stress and a change in myocardial substrate utilization (acylcarnitines transport fatty acids into the mitochondria for oxidation).

Thyroxine, a hormone with cardiomyocyte maturation and regenerative capacity[26] was significantly increased in both ICM (FC = 1.97, $P = 4.95 \times 10^{-3}$) and DCM hearts (FC = 1.7, $P = 0.04$) (Table 3, Fig. 3b,c and Supplementary Data 4–6). Biotin, a cofactor in various metabolic pathways, was significantly increased in ICM (FC = 1.8, $P = 6.7 \times 10^{-3}$) and DCM (FC = 1.5, $P = 0.04$) hearts. Likewise, guanidinoacetate, a creatine precursor, was significantly increased in ICM (FC = 1.4, $P = 0.01$) and DCM (FC = 1.4, $P = 9.73 \times 10^{-3}$) hearts. An

intermediate in glycolysis (a major source of energy in failing hearts), fructose-6-phosphate, was significantly decreased in ICM (FC = −1.67, $P = 9.6 \times 10^{-3}$) and DCM myocardium (FC = −1.67, $P = 8.3 \times 10^{-3}$), suggesting increased consumption of glycolytic intermediates in both pathologies.

Ten of the 22 DE metabolites in ICM were not regulated in DCM. These include trimethylamine-N-oxide (FC = 2.2, $P = 0.02$), known to be elevated in ischaemic heart disease[27–29], and serotonin (FC = 2.4, $P = 0.02$), a metabolite highly concentrated in platelets and released upon activation, which may reflect platelet activation in ICM. Others were deoxyribose-5-phosphate (FC = 1.2, $P = 0.04$), a pentose phosphate intermediate, known to be regulated in ischaemic heart disease[30] and citrulline (FC = 2.4, $P = 0.04$), a substrate of nitric oxide synthase.

Valine (FC = 1.6, $P = 0.006$) was significantly increased in ICM, perhaps reflecting a change in the utilization of this branched amino acid, a known cardiac substrate. cAMP, an important second messenger regulator of myocardial energetics and flow distribution in ischaemic myocardium[31], was elevated in ICM myocardium (FC = 1.3, $P = 0.04$). Some purine nucleotides were significantly different in ICM but not DCM: cytidine (FC =

**Table 4 Top 10 DE metabolites in icm vs donor myocardium.**

| Metabolite | Log2 FC | FC | t | P value | Adjusted P value |
|---|---|---|---|---|---|
| Riboflavin-5-monophosphate | −1.11 | −2.20 | −5.72 | $1.01 \times 10^{-06}$ | $1.27 \times 10^{-04}$ |
| Cytidine | 1.76 | 3.38 | 4.95 | $1.26 \times 10^{-05}$ | $5.51 \times 10^{-04}$ |
| Cytosine | 1.61 | 3.06 | 4.94 | $1.31 \times 10^{-05}$ | $5.51 \times 10^{-04}$ |
| Biotin | 0.83 | 1.77 | 4.35 | $8.46 \times 10^{-05}$ | $2.27 \times 10^{-03}$ |
| Uridine | 0.82 | 1.77 | 4.32 | $9.50 \times 10^{-05}$ | $2.27 \times 10^{-03}$ |
| 2-Deoxycytidine | 1.41 | 2.66 | 4.28 | $1.08 \times 10^{-04}$ | $2.27 \times 10^{-03}$ |
| Thyroxine | 0.98 | 1.97 | 3.97 | $2.75 \times 10^{-04}$ | $4.95 \times 10^{-03}$ |
| Adenosine | 0.83 | 1.78 | 3.93 | $3.17 \times 10^{-04}$ | $4.99 \times 10^{-03}$ |
| Valine | 0.65 | 1.57 | 3.83 | $4.21 \times 10^{-04}$ | $5.90 \times 10^{-03}$ |
| Fructose-6-phosphate | −0.73 | −1.67 | −3.63 | $7.65 \times 10^{-04}$ | $9.64 \times 10^{-03}$ |

Estimates were derived using the linear regression model adjusted for gender, where $n_{ICM} = 15$ and $n_{donor} = 15$; P values were adjusted for multiple comparisons by the Benjamini–Hochberg method. (See Supplementary Data 4).
FC fold change, t t-statistic.

**Table 5 Top 10 DE Metabolites in DCM vs Donor myocardium.**

| Metabolite | Log2 FC | FC | t | P value | Adjusted P value |
|---|---|---|---|---|---|
| Riboflavin-5-monophosphate | −1.27 | −2.44 | −6.44 | $9.57 \times 10^{-08}$ | $1.21 \times 10^{-05}$ |
| Kynurenine | 0.59 | 1.51 | 4.37 | $8.04 \times 10^{-05}$ | $5.06 \times 10^{-03}$ |
| Fructose-6-phosphate | −0.84 | −1.78 | −4.08 | $1.96 \times 10^{-04}$ | $8.25 \times 10^{-03}$ |
| Cystamine | 1.17 | 2.26 | 3.88 | $3.64 \times 10^{-04}$ | $9.73 \times 10^{-03}$ |
| Guanidoacetate | 0.52 | 1.43 | 3.86 | $3.86 \times 10^{-04}$ | $9.73 \times 10^{-03}$ |
| Ureidopropionic-acid | −0.36 | −1.28 | −3.72 | $5.80 \times 10^{-04}$ | $1.20 \times 10^{-02}$ |
| N-Acetyl-L-alanine | −0.33 | −1.25 | −3.68 | $6.66 \times 10^{-04}$ | $1.20 \times 10^{-02}$ |
| 2-Methylacetoacetate | 0.40 | 1.32 | 3.51 | $1.09 \times 10^{-03}$ | $1.58 \times 10^{-02}$ |
| Butyrlcarnitine | −0.21 | −1.16 | −3.50 | $1.13 \times 10^{-03}$ | $1.58 \times 10^{-02}$ |
| 2-Methylbutyrylcarnitine | −3.33 | −10 | −3.41 | $1.44 \times 10^{-03}$ | $1.82 \times 10^{-02}$ |

Estimates were derived using the linear regression model adjusted for gender, $n_{DCM} = 14$ and $n_{donor} = 15$; p values were adjusted for multiple comparisons by the Benjamini–Hochberg method. (See Supplementary Data 5).
FC fold change, t t-statistic.

3.4, $P = 5.5 \times 10^{-4}$), cytosine (FC = 3.1, $P = 5.5 \times 10^{-4}$), and deoxycytidine (FC = 1.4, $P = 2.3 \times 10^{-3}$). However, the purine nucleotide n-acetyl-l-alanine was decreased in both ICM (FC = −1.22, P = 0.02) and DCM (FC = −1.25, P = 0.01). A pyrimidine nucleotide, uridine, was increased in both ICM (FC = 1.8, P = $2.3 \times 10^{-3}$) and DCM (FC = 1.5, P = 0.04).

**Pathway analyses.** Pathway analysis was performed at both the protein and metabolite levels for both ICM and DCM with annotation by KEGG[32] (Fig. 4). Figure 4a displays the common pathways regulated in ICM and DCM at both the protein and metabolite levels. In ICM and DCM, there were 9 and 10 pathways, respectively, regulated at both the protein and metabolite level. In both diseases, thyroid hormone synthesis, carbohydrate digestion and absorption, estrogen signalling pathway, "metabolite pathways", and aldosterone regulated sodium reabsorption pathways were regulated at both the protein and metabolite levels revealing a core set of pathways regulated in advanced heart failure.

In Fig. 4b, we illustrate the protein and metabolite changes in the Estrogen signalling pathway as an example of pathway level changes. As illustrated in Fig. 4b, the pathway analysis revealed the majority of proteins associated with the estrogen signalling pathway were co-regulated in both ICM and DCM. However, our data also revealed specific differences between the diseases. For example, CALM2, a calcium binding protein, was upregulated in ICM and downregulated in DCM. The upregulation in ICM may be related to the calcium overload and altered calcium cycling into the endoplasmic reticulum post ischaemia[33]. Supplementary

Data 7-10 contain all the regulated pathways at the protein and metabolite levels in ICM and DCM.

One of the most significant protein pathways in both ICM ($P = 4.3 \times 10^{-4}$) and DCM ($P = 2.7 \times 10^{-18}$) was the oxidative phosphorylation pathway, highlighting the central role of this canonical energetic pathway (see Supplementary Data 7, 8 for a full list of protein pathways in ICM and DCM). Complement and coagulation cascade pathway was also amongst the most significant pathways in both ICM ($P = 1.2 \times 10^{-8}$) and DCM ($P = 4.6 \times 10^{-10}$) at the protein level. Key metabolic substrate pathways such as glycolysis (ICM: $P = 1.4 \times 10^{-3}$; DCM: $P = 7.5 \times 10^{-3}$) were also amongst the top ranked pathways. The TCA cycle ($P = 6.1 \times 10^{-3}$), carbohydrate metabolism ($P = 3 \times 10^{-3}$), and pyruvate ($P = 0.026$) metabolic pathways were enriched in ICM but not DCM. Conversely, fatty acid elongation / lipogenesis pathway ($P = 9.5 \times 10^{-3}$), cholesterol metabolism ($P = 0.018$), and fat digestion ($P = 0.03$) were significant in DCM but not ICM. There was less overlap in metabolite pathways in ICM and DCM (see Supplementary Data 9, 10 for a full list of metabolite pathway rankings in ICM and DCM).

**Network analyses.** We next performed pair-wise correlation network analysis of all proteins and metabolites to identify co-regulated nodes. Figure 5a, b show network analyses highlighting the most correlated metabolites and proteins in ICM and DCM. Each node represents a DE protein (circle) or a DE metabolite (square), where the size of each node is proportional to its eigenvector centrality (network influence), and nodes that are more correlated are plotted closer to each other. An edge exists

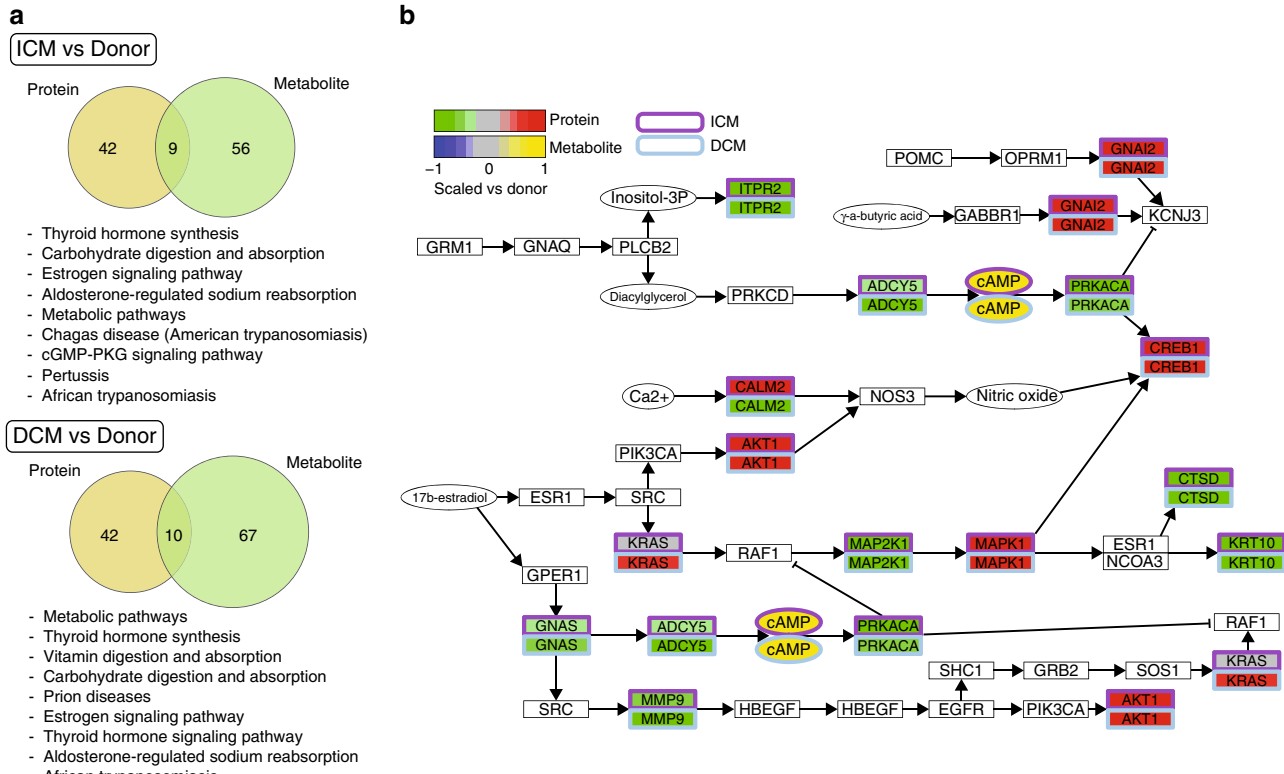

**Fig. 4 Pathways analysis. a** Venn diagrams illustrating the number of metabolite (yellow) and protein (green) pathways significantly changed in ICM and DCM, followed by list of pathways in ICM and DCM that were regulated at both the protein and metabolite level. **b** Illustrative example of protein and metabolite changes in Estrogen pathway. Source data are provided as a Source Data file.

when it connects two nodes with a Pearson correlation coefficient greater than or equal to 0.7 based on their abundance levels (blue are positive correlations and red are negative correlations), with edge width proportional to Pearson correlation coefficient. Clustering generally represents grouping by function, and co-clustering of metabolites with proteins illustrates their relationship in a common pathway. For example, transthyretin, a transport protein that carries thyroxine, is most closely related to thyroxine. To aid visualisation, we annotated the functional classification of each protein node using DAVID[34,35].

In ICM (Fig. 5a) there was a closely correlated group (in purple) of large extracellular matrix nodes (denoting large network influence), indicating their importance in myocardial remodelling in ICM. There was also a closely correlated group of complement and coagulation nodes (light green/blue), and several cytoskeleton (light orange) and immune (yellow) nodes that are less co-regulated as a group. Whilst the cytoskeleton nodes did not cluster tightly, their size indicates that they do exert significant network influence. In DCM (Fig. 5b), there were many co-regulated extracellular matrix proteins (purple) that accounted for the majority of the nodes, underlining the importance of extracellular remodelling in this disease. There was also a significant complement/coagulation (light green/blue) component to the network, with a smaller number of cytoskeleton (light orange) and immune (yellow) nodes than in ICM.

**Gender-specific ICM and DCM DE proteins and metabolites.** As there are gender-specific differences in cardiovascular disease presentation, outcomes, response to treatment[36], and type of heart failure[37], we next examined whether protein and metabolite gender-specific differences were informative. First, in an analysis of differential proteins by gender *independent* of disease, the only

differential protein was EIF1AY (Eukaryotic translation initiation factor 1 A, Y-chromosomal), a protein encoded by a gene on the Y-chromosome. In the same analysis, there were no differential metabolites. We next examined gender interactions in ICM and DCM.

In ICM vs donor, there were 18 proteins with a gender interaction effect. In both ICM (FC = −200, P = 9.5 × 10⁻⁶) and DCM (FC = −3,333, P = 1.9 × 10⁻⁹), protein AP1M1 was dramatically reduced in male hearts (Fig. 5e, Supplementary Data 11, 12). Protein FRY was significantly reduced (FC = −76, P = 0.002), whereas X-chromosome-encoded protein HNRNPH2 (FC = 6.96, P = 0.035) was significantly elevated in male ICM hearts only. In ICM vs donor, there were 14 DE metabolites with a gender interaction effect (Supplementary Data 13). Several of these were also regulated in DCM hearts. For example, the top metabolite with gender interaction in ICM (FC = 4.1, P = 2.5 × 10⁻⁹) was histidine, increased in male hearts. This was also the top metabolite with gender interaction in DCM, similarly increased in male hearts (FC = 4.2, P = 8.8 × 10⁻¹⁰). The reason for this is unclear, although there has been a report of gender-specific anorexia induced by histidine[38]. Methylbutyrylcarnitine, a mitochondrial substrate, was significantly decreased in male ICM (FC = 2.8, P = 9.8 × 10⁻⁴) and DCM (FC = 3.1, P = 2.4 × 10⁻⁸) hearts. Ornithine, a substrate of X-chromosome-encoded ornithine transcarbamylase, was significantly elevated in male ICM (FC = 2.8, P = 0.004) and DCM (FC = 2.3, P = 0.03) hearts. In ICM only, cysteamine (FC = 2.5, P = 0.004), spermine (FC = 2.3, P = 0.006), 2-deoxycytidine (FC = 2.4, P = 0.02), tryptophan (FC = 2.1, P = 0.02), cytosine (FC = 2.5, P = 0.02), tyrosine (FC = 2.3, P = 0.02), ADMA (FC = 2.0, P = 0.03), and p-aminobenzoate (FC = 1.7, P = 0.03) were increased in male hearts, whereas ATP (FC = −3.33, P = 0.02) was decreased in

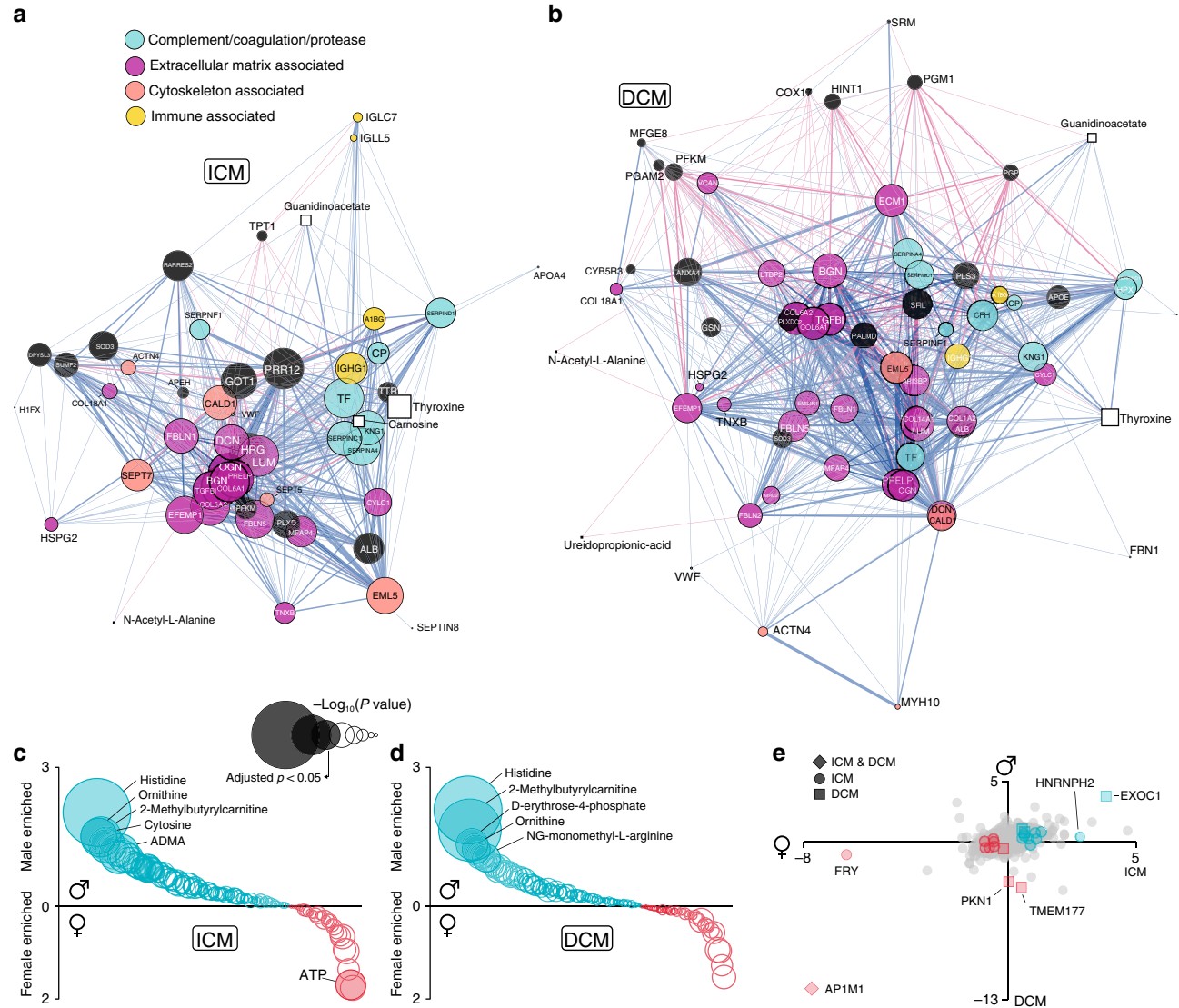

**Fig. 5 Network analysis. a**, **b** Network analyses illustrating functional protein (circles) and metabolite (squares) nodes in ICM and DCM left ventricular myocardium. Nodes colour coded according to functional classification. Plot illustrating metabolite-gender interaction in ICM (**c**) and DCM (**d**), and protein-gender interaction in ICM and DCM (**e**). Source data are provided as a Source Data file.

male hearts. In DCM vs donor, there were 7 proteins with gender interaction (Supplementary Data 14). EXOC1, a protein necessary for exocytic vesicle targeting, was dramatically elevated in male DCM hearts (FC = 13.5, $P = 4.1 \times 10^{-7}$). ANXA3, silencing of which leads to improved myocardial repair post myocardial infarction[39] and shown to have a role in early blood vessel formation[40], was reduced in male DCM hearts (FC = −1.42, $P =$ 0.03). In DCM vs donor, there were 7 metabolites with gender interaction, including an increase of NG-monomethyl-L-arginine, an inhibitor of nitric oxide synthase in male hearts (FC = 2.1, $P = 0.04$). Although only nominally significant in the gender-heart failure interaction analysis ($P = 0.014$), in a separate analysis within males and females, TMAO was elevated in *male* ICM (FC = 1.95, adjusted $p$ value = 0.018) and DCM (FC = 1.96, adjusted $p$ value = 0.026) compared to male donor hearts, whereas it was not significantly different between female HF and female donor hearts.

## Discussion

In ICM and DCM human hearts, compared to age-, gender, and BMI-matched donors, we have identified key disease pathways, in

addition to creating a proteomic-metabolomic resource for heart failure researchers. We confirmed established pathogenesis in ICM and DCM including changes in many ECM proteins, a well-established feature of cardiac remodelling[41], and in ICM changes in von Willebrand factor and serotonin indicating platelet activation, along with components of the blood coagulation cascade, highlighting the atherothrombotic pathogenesis of ICM. In ICM hearts, serum amyloid A was decreased 10-fold, the first report of this protein perturbation in human ICM myocardium. It is not clear why an acute-phase protein stimulated by inflammation and reported to be positively associated with coronary artery disease[42,43] was dramatically decreased. However, it is important to highlight that it was measured in chronic, rather than acute, inflammation, and in advanced heart failure. Supportive of this finding, Chen et al. recently determined that SAA1 gene expression was significantly reduced in ICM myocardium and SAA1 protein differentially expressed between DCM, ICM, and hypertrophic cardiomyopathy[17].

A key perturbed metabolite, common to both ICM and DCM, was thyroxine. As none of the ICM or DCM patients had systemic thyroid dysfunction, this may indicate that thyroid

hormone plays a central and ongoing role in normal cardiac function, as it was perturbed in HF hearts regardless of aetiology. Patients with HF often have a condition known as "low T3 syndrome", where lower plasma levels of the active triiodothyroxine or T3, produced from thyroxine (T4), mean less availability to the heart. Inside failing cardiomyocytes, less T3 is available because of lower D2 deiodinase activity, causing less T3 to be converted from T4, but probably also because of increased D3 deiodinase activity causing inactivation of T3 (ref. [44]). The consequences of this syndrome are incompletely understood[45,46]. Treatment with levo-thyroxine or T4 increases the T4/T3 ratio and exacerbates low T3 in the heart[44], but it is unclear if this is the underlying mechanism for increased mortality in HF patients treated with levo-thyroxine[44]. Recent work by Gil-Cayuel et al. demonstrated altered expression of two genes in thyroid hormone biogenesis and reduction of T3 in ischaemic myocardium[47]. The same group also demonstrated altered expression of the same genes in DCM myocardium, along with increased T4 and decreased T3 (ref. [48]). Our work buttresses these findings by directly showing increased T4 in ICM and DCM left ventricular myocardium. It has also been suggested that the changes in local cardiac conversion of T4 to T3 are less relevant than the reduced plasma T3 levels in heart failure, as the majority of cardiac T3 comes from the circulation, with only up to 7% resulting from conversion of T4 by D2 deiodinase in the heart[49]. However, our data is supportive of the Gil-Cayuela et al. data[47,48], and together these data suggest that in heart failure myocardium, low cardiac T3 coming from the circulation is likely exacerbated by reduced local conversion by less active D2 deiodinase, which led to the elevated T4 we saw in the myocardium. Hence, a localised (myocardial) thyroid hormone regulatory system in this context has important implications for developing heart failure therapeutics.

The most significantly regulated metabolite in both ICM and DCM was FMN. This oxidoreductase cofactor and component of electron transport complex I in mitochondria was ~2-fold decreased in both cardiomyopathies, perhaps indicative of increased oxidative stress in advanced HF, regardless of aetiology. Whilst decreased FMN has been reported in canine myocardium post ischaemia[50], this is the first report to our knowledge of changes in this cofactor in human heart failure myocardium. Furthermore, it has also been demonstrated in rat hearts that administration of FMN, along with other riboflavin derivatives, is protective post ischaemia-reperfusion via marked recovery of high energy phosphate compounds and pH in the extracellular space[51]. Administration of related metabolite cytoflavin was shown to be cardioprotective in doxorubicin-induced cardiomyopathy in rats[52]. Therefore, there is pre-clinical evidence that FMN is reduced in ischaemic myocardium, and that administration of FMN and/or related metabolites is cardioprotective in cardiac ischaemia and toxic cardiomyopathy[50–53]. However, we are the first to show a decrease in FMN in human heart failure myocardium. In fact, FMN had the greatest fold change and significance of all metabolites in both ICM and DCM myocardium; whether it is a viable therapeutic target in the treatment of these two most common human heart failure aetiologies remains to be seen.

In both ICM and DCM, pyrimidine and purine nucleotides were significantly changed. This has been described in the circulation of HF patients before[54], and suggested to result from an increased switch from aerobic to anaerobic glycolysis, which provides less ATP, leading to increased catabolism of purines. Here we show these changes in the myocardium itself, confirming that these peripheral changes in the circulation of heart failure patients were most likely derived from perturbations of these substrates in the heart. Reported changes in glycolysis and glucose

oxidation are inconsistent, dependent on stage of HF and aetiology[55]. In both ICM and DCM hearts, fructose-6-phosphate, a glycolytic intermediate, was significantly decreased 2-fold, suggesting increased consumption of glycolytic intermediates and utilization of this pathway for energy generation in HF hearts, and consistent with previous reports of increased glucose utilization in patients with idiopathic DCM[56].

Our analysis also informs observations made previously at the transcript and protein level. For example, recent work by Chen et al.[13] found APOA4 mRNA to be upregulated in DCM hearts. We importantly confirm that this regulation of APOA4 is maintained at the protein level. Furthermore, our data validates other protein expression changes in this paper, such as the upregulation of proteins PRELP and MFAP4 in ICM myocardium, along with members of the NDUF, TF, and COL families of proteins in both ICM and DCM myocardium[13]. There were also many differences with the Chen et al. study[13], for example the thyroid hormone pathway; this is likely because our metabolite data captured the myocardial increase in T4; the lack of proteomic changes in both ours and the Chen et al. dataset likely represents the difficulties measuring deiodinase proteins using standard proteomic extraction and analytic techniques.

Recent work has highlighted the differences in gene expression between male and female hearts[57], with several exploring the interaction of gender with heart failure[58–61]. Using left ventricular myocardium derived from explanted hearts or obtained at the time of left ventricular assist device insertion, Fermin et al. explored cardiac sexual dimorphism at the gene transcript level and reported >1800 genes showing sexual dimorphism in the heart, many involved in ion transport or G-coupled receptor signaling[58]. Using ventricular myocardium from 21 failing compared to 4 non-failing hearts at transplant, Boheler et al. found significant interactions between heart failure and gender, and transcript changes that were specific to gender, e.g. Gpd58 decreased in male HF but increased in female HF[59]. Kararigas et al. found 36 17β-estradiol E2-dependent genes regulated in a sex-specific manner[60], after exposing epi-myocardium obtained at the time of coronary artery bypass grafting to E2 or hydroxypropyl-β-cyclodextrin. In both ICM and DCM myocardium, Y-chromosome related protein E1F1AY was dramatically elevated in male vs female heart failure myocardium, having previously been reported at the transcript level only[61]. In our study, we saw several novel gender interactions at both the protein *and* metabolite level. We determined several mitochondrial-related gender interactions: e.g. FMN, which is a co-factor in mitochondrial oxidoreductase reactions, and mitochondrial substrates like short-chain acylcarnitines, all consistent with the gender-specificity of mitochondrial function[62,63]. Notably, inherited short-chain acylcarnitine disorders, e.g. carnitine palmitoyl transferase II deficiency, are inherited in an autosomal recessive manner, but predominantly affect males. Ornithine, metabolized by X-chromosome-encoded ornithine transcarbamylase, had a significant gender interaction. A divergent effect of androgens on ornithine decarboxylase activity in rat hearts has been reported before[64], but to our knowledge we are the first to report ornithine's interaction with gender in human heart failure myocardium. TMAO was significantly elevated in male heart failure myocardium only, consistent with its metabolism by gender-specific flavin-containing monooxygenases[65]. We are the first to report the interaction of TMAO with gender in human heart failure myocardium, which has important implications when considering that this microbiome-derived metabolite has been reported widely as a mediator of atherosclerosis, platelet activation, and myocardial infarction[27–29,65]; our data suggests it may be a contributor to the divergence in heart failure outcomes in men and women. An interaction of nitric oxide deficiency with

gender in recovery post ischaemia has been reported before in mice[66], and S-nitrosoglutathione reductase was reported to be a critical sex-dependent mediator of myocardial protein S-nitrosylation in mice[67] and murine ex vivo hearts[68] (more active in females); S-nitrosylation is reported as an essential mediator of nitric oxide-dependent cardioprotection[67]. Other work in mice revealed that nitroglycerin-induced calcitonin gene-related peptide release is eNOS-dependent, with a greater response in females[69], and estrogen has been shown to upregulate NOS in neonatal rat cardiomyocytes[70]. However, we are the first to report that in both ICM and DCM, inhibitors of nitric oxide synthase (ADMA and L-NMMA, respectively) were increased in male hearts only, implicating perturbed NOS activity as a predominant driver of male, but not female, HF. Therefore, we report novel insights into the interaction of gender with heart failure, and extended observations only seen previously at the transcript level or in animal studies.

This is the first study to apply comprehensive "downstream" 'omic screening to a large cohort of cryopreserved, human ICM and DCM left ventricular myocardium with appropriate matching to histopathologically-normal donor hearts. We report perturbations in well-established pathological pathways in heart failure myocardium such as fibrotic remodeling and oxidative stress, and confirm previous perturbations in local cardiac thyroid metabolism. Many of the perturbed pathways were common to both heart failure aetiologies, revealing overlapping molecular changes as heart failure advances. We report novel changes in ICM and DCM myocardium, including serum amyloid A1 protein, FMN, purines and pyrimidines, and novel gender-specific perturbations in histidine, acylcarnitines, ornithine, microbiome-derived atherogenic factor TMAO, and nitric oxide metabolites, permitting a better understanding of divergent HF pathogenesis between males and females. Our networks succinctly illustrate the most influential nodes in both types of heart failure; for example, in ICM myocardium, a cluster of ECM proteins such as EFEMP1, LUM, and COL6A2 are highly influential, as is TF in the coagulation cascade. Beyond serving as a rich resource, our study provides several novel pathway changes with therapeutic implications. To enable the broader research community, we have created an interactive online repository (https://mengboli.shinyapps.io/heartomics/) as a publicly-available resource.

## Methods

**Human heart tissue**. Donor hearts were procured but not used for heart transplantation (reasons included transportation logistics, donor-recipient mismatch in size, and immune incompatibility), whilst heart failure samples comprised those patients undergoing heart transplantation; both cohorts as previously described[1,2,4–8,71,72]. These samples are not post-mortem. Histological analysis of the donor samples were shown to be structurally normal as per formal pathological examination[2,3]. LV samples from both donors and heart failure patients were collected and and snap frozen (−196 °C) immediately within the operating theatre. All samples were stored in the Sydney Heart Bank at −170 to −180 °C from the time of harvest[1]. Written consent was obtained to use tissue specimens for research. The study was approved by the Ethics Committee of The University of Sydney (USYD #2016/923) and was conducted in accordance with the Declaration of Helsinki.

**Metabolomics**. The mass spectrometry metabolomics data have been deposited to Metabolomics workbench. Frozen heart tissues were crushed to powder in liquid nitrogen and weighed for metabolomic (~150 mg) and proteomic (~20 mg) analysis, respectively. For targeted metabolomic analysis, a tandem liquid chromatography–mass spectrometry (LC-MS/MS) composed of an Agilent (Santa Clara, CA, USA) 1260 Infinity liquid chromatography system coupled to a QTRAP5500 mass spectrometer (AB Sciex, Foster City, CA, USA) was used to measure hydrophilic metabolites in positive and negative ionization modes. The polar metabolites in positive ionization mode included nucleotides, neurotransmitters, amino acids and vitamins, and were separated on a hydrophilic interaction chromatography using acidic chromatographic condition. For the separation of high-energy intermediates, nucleotide and nucleoside phosphates, organic acids, Krebs cycle intermediates and glycolytic intermediates, a BEH Amide

column was used, both as previously described[73]. Both sets of metabolites were based on HPLC-grade authentic chemical standards, where the retention time (RT) and MRM Q1 and Q3 masses were unique to each metabolite were derived and compiled into a library. A data file, "Instrument Settings Metabolite", containing the LC/MS instrument settings for each metabolite, has been uploaded to Supplementary Data 15.

The analysis software MultiQuant 3.0 (ABSciex) was used for MRM Q1/Q3 peak integration of the raw data files (Analyst software, v.1.6.2; ABSciex). The peak area corresponds to the abundance of that metabolite; the area values were then normalized to their bookended pooled plasma samples, which were included after every 4 study samples in the sample queue, in order to account for any performance drift in the LC-MS/MS.

**Proteomics**. The mass spectrometry proteomics data have been deposited to the ProteomeXchange Consortium via the PRIDE partner repository with the dataset identifier PXD018678. Muscle was lysed by tip-probe sonication in 8 M guanidine containing 100 mM Tris (pH 8.0), 10 mM tris(2-carboxyethyl)phosphine, 40 mM 2-chloroacetamide and heated to 95 °C for 5 min. The lysate was centrifuged at $20,000 \times g$ for 20 min at 4 °C and the supernatant diluted 1:1 with water. Proteins were precipitated with 4 volumes of acetone overnight at −30 °C and washed with 80% acetone. Proteins were digested in 100 mM Tris pH 7.5 containing 10% 2,2,2-Trifluoroethanol overnight at 37 °C with sequencing grade LysC (Wako Chemicals) and trypsin (Sigma) at an enzyme to substrate ratio of 1:50. The digests were acidified to 1% trifluoroacetic acid prior to purification by styrene divinyl benzene—reversed phase sulfonated solid phase extraction microcolumns. Peptides were spiked with iRT peptides (Biognosys) and analysed with a Dionex 3500 ultra-high performance liquid chromatography (UHPLC) coupled to a Q-Exactive HF-X in positive polarity mode. The peptides were separated on an in-house 100 μm x 15 cm column with an integrated emitter using a Sutter laser puller (1.9 μm particle size, C18AQ; Dr Maisch) with a gradient of 2 – 35% acetonitrile containing 0.1% formic acid over 60 min at 800 nl/min. Data were acquired with positive ionization and the instrument was operated in data-independent acquisition mode (DIA) using settings essentially described previously[74]. Briefly, an MS1 scan was acquired from 350 – 1650 m/z (30,000 resolution, 3e6 AGC, 50 ms injection time) followed by 20 variable window sized DIA isolations and fragmentation with HCD (15,000 resolution, 1e5 AGC, 3e6 AGC, automatic injection and step normalized collision energies of 22.5, 25 and 27.5).

A spectral library was created by fractionating a pooled mix of peptides from all the samples on an in-house packed 320 μm × 25 cm column (3 μm particle size, BEH; Waters) with a gradient of 2–40% acetonitrile containing 10 mM ammonium formate over 60 min at 6 μl/min using an Agilent 1260 HPLC. A total of 12 concatenated fractions were analysed using the identical gradient conditions above except the instrument was operated in data-dependent acquisition (DDA) mode. Briefly, an MS1 scan was acquired from 350 – 1650 m/z (60,000 resolution, 3e6 AGC, 50 ms injection time) followed by 20 MS/MS with HCD (1.2 m/z isolation, 15,000 resolution, 1e5 AGC, 27 NCE). DDA data were processed with Andromeda in MaxQuant v1.6.0.9 against the human UniProt database (September, 2017) using all default settings with peptide spectral matches and protein false discovery rate (FDR) set to 1%[75]. First search mass tolerances were set to 20 ppm for MS1 and MS2 and following recalibration, a second search was performed with MS1 tolerance of 4.5 ppm and MS2 tolerance of 20 ppm. The MaxQuant output were used to generate a spectral library in Spectronaut v12 and the DIA data were processed with default settings employing retention time and mass recalibration. Quantification was performed using MS2-based extracted ion chromatograms employing 3-6 fragment ions >450 m/z with automated fragment-ion interference removal as described previously[76]. Data was post-processed via Log2 transformation and median normalization in Perseus[77].

**Statistics**. All statistical analyses were then done in R (3.5.2). Both metabolomic and proteomic data were Log2 transformed and median normalized. Differential expression (DE) analyses for both proteins and metabolites by disease groups (ICM vs Donor and DCM vs Donor) adjusted for gender were performed using the lmFit and eBayes functions implemented in the limma R package[78], where a linear model is fitted for each protein or metabolite to identify those that are differentially expressed in each disease group compared with the healthy donors while adjusting for gender. We used multi-group comparison in design of limma, that is, all samples from ICM, DCM and donors were used to estimate the variance of each protein or metabolite. We then performed the contrasts between the two groups of interest to report the direct differential expression in proteins or metabolites for ICM vs Donor or DCM vs Donor. Pathway analyses were performed with mean-rank gene set tests, using the geneSetTest function from limma, where t-statistics from the corresponding differential expression analysis were ranked. Pathways were annotated by the Kyoto Encyclopedia of Genes and Genomes[32] (https://www.kegg.jp). Pathway illustrations were generated using the pathview R package[79]. Gender analysis was performed by fitting interaction models to look for any protein and metabolite that were differentially perturbed in each gender. Network plots of DE proteins and metabolites were generated with the igraph R package[80]. Nodes in the network plots are the differentially expressed (DE) proteins in direct comparisons (ICM vs Donor and DCM vs Donor) while adjusted for gender with an adjusted P value ≤ 0.01 and DE metabolites with an adjusted P value ≤ 0.05 for

ICM vs Donor, and for DCM vs Donor, nodes are the DE proteins with an adjusted $P$ value $\leq 0.001$ and DE metabolites with an adjusted $P$ value $\leq 0.05$. Edge weights were calculated as pairwise Pearson correlation coefficient for every two nodes, and only edges with an absolute weight greater than or equal to 0.7 were shown in the plots. The widths of edges in the plots are proportional to the corresponding edge weights. The graph layout was by multidimensional scaling, where the distance between two nodes is proportional to the inverse of their pairwise Pearson correlation coefficient. The Benjamini–Hochberg method was used to calculate adjusted $P$ values for multiple comparisons in all mentioned analyses. Functional annotation of nodes in each network that are proteins were performed using DAVID[34,35].

**Reporting summary.** Further information on research design is available in the Nature Research Reporting Summary linked to this article.

## Data availability

All relevant data are available from the authors. All data generated in this study are available in public repositories. The mass spectrometry metabolomics data have been deposited to Metabolomics workbench with the Project ID PR000928 and Study ID ST001364. The mass spectrometry proteomics data have been deposited to the ProteomeXchange Consortium via the PRIDE partner repository with the dataset identifier PXD018678. The source data underlying all presented figures and tables are provided as a Source Data file.

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

## Acknowledgements

We thank the patients and staff of St Vincent's Hospital, Sydney, and the Australian Red Cross Blood Service. We thank Emeritus Prof. Cris dos Remedios and the Late Dr. Victor Chang AC. We thank Ben Crossett and Stuart Cordwell from Sydney Mass Spectrometry. This was supported by the National Health and Medical Research Council (NHMRC) of Australia. The contents of the published material are solely the responsibility of the individual authors and do not reflect the view of NHMRC.

## Author contributions

M.L. jointly conceived the study; performed the statistical analyses; created the figures and the online interactive application; and wrote the manuscript. B.L.P. performed and analysed the proteomics analysis, created the figures, and wrote the manuscript. E.P., B. H., J.C., and Y.C.K., and O.G. retrieved the samples, processed the tissue, and extracted the samples for metabolomics. Y.C.K. designed, performed, and analysed the metabolomics study. D.E.J. supervised and provided intellectual critique of the proteomics data and the manuscript. J.Y. supervised the statistical analyses. S.L. jointly conceived the study, oversaw the myocardial tissue retrieval and study design, and wrote the manuscript. J.F.O'S. conceived the study, oversaw the metabolomics analysis, and wrote the manuscript.

## Competing interests

The authors declare no competing interests.
