## [Peer Review File · Nature Communications]

Reviewers' comments:

Reviewer #2 (Remarks to the Author):

Summary

The manuscript describes proteomic and metabolomic datasets from patients with ischemic and dilated cardiomyopathy relative to controls. There is a paucity of high-quality human heart failure proteomic data in the literature, and existing datasets have often been used in furtherance of specific hypotheses, largely relegated to supplements. Human HF proteomes, therefore, remain undercharacterized, particularly as it pertains to etiological and sex differences. Any study that addresses these lucidly would be a welcome advance.

Evaluation Overview

I have little doubt that the authors are sitting on a treasure trove of data. My primary disappointment is that the data is poorly showcased by the manuscript, which appears to be written as a short communique. Notwithstanding the potential impact of the human datasets, the manuscript, as currently laid out and written, would encounter headwinds at a middling proteomics journal, as the data presentation is quite awkward and opaque. This not simply an issue of “cosmetics”, as I understand that researchers can reasonably differ on how to present complex data sets. There are also quality control novelty issues. Therefore, I cannot recommend the manuscript for publication in its present form.

Strength: The authors have provided a link that allows the reader to interact with the dataset and observe pathways and networks. The reviewer appreciates the care taken with the layout of the website at <https://mengboli.shinyapps.io/heartomics/>.

Major Issues.

1. Notwithstanding that the journal is called Nature Communications, the choice of reporting the study as a short communique, rather than a regular research article, is problematic. The lay of the research landscape is not adequately framed, the data itself is given short shrift in the figures and the text is highly descriptive rather than quantitative. Essentially the figures themselves are more metadata than actual data (particularly fig 2). The data is confined largely to the supplements. Finally, posited new findings are not adequately contextualized, given that there are likely hundreds of cardiac profiling studies that have preceded this one.

2. The schematic in figure 1, does not do either the data workflow or the patient cohort justice. Firstly, the nature of the patient cohort should be addressed separately in a more detailed table. More on that later. The proteomic and metabolomic workflow is highly generic. The pictogram essentially says “we did proteomics and metabolomics”. It does not say what kind of proteomics (label-free? TMT?) or metabolomics (LC? GC?, a combination?). The green balloon says “Data processing & data analysis”. It does not say what kind of analysis (PCA? Statistical tests? Pathways? Networks?). If the point is to encapsulate the meat of the study in one glance, it could be greatly improved.

3. On to patients. Figure 1b and 1c are insufficient documentation of the patient cohort. Ideally, this would be done in two ways. The manuscript would benefit from a table summarizing the patient vital statistics (height, weight, sex, race) but most importantly, documenting the cardiac functional parameters that led to their inclusion. What was the ejection fraction? A supplement containing detailed table documenting each patient, with vital statistics, height, age, weight, sex, ejection fraction, fractional shortening, left ventricle internal diameter, drug regimen, smoking history, etc. would also be useful.

4. These major characteristics of the cohort should be incorporated into the text.

5. I took a look at the dataset submission. The authors have submitted the raw data files. They have also submitted maxquant archive with all the pertinent peptide and protein particulars. What is lacking (this is common to many submissions) is a simple table of protein quantitation (e.g. sample # vs protein accession), essentially equivalent to a GEO matrix file that any user can open in excel or any other data analysis program. This is key if the dataset is to be mined as a resource to the broader community and not gather dust.

6. The proteomic and metabolomic methods could use some fleshing out. As written, it is not clear how the metabolites were identified. MRM yes, but was this against a previously compiled database of standards? a search? What were the instrument settings, tolerances, etc.?

For the proteomics section what were the MS1 and MS2 mass tolerances. It's not clear what is being quantified. If the authors are using Maxquant, they using iBAQ or LFQ intensities?

7. Particularly in the proteome, is there any missing data? What was its distribution? How much missing data would prompt exclusion of protein from the analysis? Was missing data imputed? If so how?

8. There is no 10,000 ft view or data QC in the form of a principal component analysis (PCA). How noisy are the data sets? How different are the normal, ICM and DCM proteomes and metabolomes. Does diagnosis track with the first principal component?

9. The authors are to be commended for going with a statistical framework like Limma which increases power. But they have chosen to go with binary comparisons (Limma modified t-test) and volcano plots, when the experimental design screams for a Limma/ANOVA multigroup comparison. This would maximize power by providing more data points for modeling. More importantly, though, it would allow the authors use an intuitive p-value-filtered, hierarchically-clustered heatmap that would showcase clusters defining the similarities and differences between normal, ICM and DCM groups. This would be a substantial improvement over the Venn diagrams in figure 2a, which are more metadata than data. Volcano plots are actual data but don't show the relationships between ICM and DCM. For instance, in the current analysis there are 135 differentially expressed proteins shared between ICM vs. Donor and DCM vs Donor. Are they regulated in the same direction for each etiology? Presumably a mix of up and down. In a Multigroup LIMMA/ANOVA-heatmap layout that would jump out. The same goes for the metabolites

10. Figure 2b is a highly uninformative display of metadata, based on the supplements. It shows the overlap between protein and metabolite pathways. Indeed integration of metabolite and protein data is difficult, but there are more sophisticated publically available tools that would likely help the authors substantially. Such tools allow you to load protein and/or gene, and metabolite levels and perform combined pathway enrichment and topological analysis. An example of this would be MetaboAnalyst. Where metabolite and protein data corroborate each other inference is strengthened. If performed downstream of cluster analysis as described above, the insights could be great.

11. Or at the very minimum, the authors could find a way to present the tabular data in their supplements graphically in the manuscript, even if it's as simple as a bar graph ranked by $-\log(p\text{-value})$

12. Because the authors have eschewed the traditional article format they do not adequately contextualize the significant body of established work on the link between both circulating and local cardiac levels of thyroid hormone and heart failure. This would include (but not be limited to) work suggesting a local cardiac T3 deficiency. E.g. the PNAS paper from several years ago showing that cardiac overexpression of the deiodinase rescues cardiac function while not affecting hypertrophy in a mouse TAC model.

Trivieri, M.G., Oudit, G.Y., Sah, R., Kerfant, B.G., Sun, H., Gramolini, A.O., Pan, Y., Wickenden, A.D., Croteau, W., Morreale de Escobar, G., et al. (2006). Cardiac-specific elevations in thyroid hormone enhance contractility and prevent pressure overload-induced cardiac dysfunction. *Proceedings of the National Academy of Sciences of the United States of America* 103, 6043-6048.

<https://www.pnas.org/content/103/15/6043>

This reviewer is not affiliated in any way. So high cardiac T4 in the authors' study might be consistent with a bottleneck caused by low cardiac de-iodinase activity. It is up to the authors to make the case for the novelty of their finding.

13. In figure 3. The authors have performed network analysis on correlated proteins and metabolites to find co-regulated nodes. The networks, as presented, are OK, but again there is perhaps a missed opportunity really chunk down the data and present it in a graphically intuitive manner. First, there is nothing wrong with a correlation network, but we now all have access the latest StringdB databases <https://string-db.org/> where the authors could build functional association networks based on gene correlation, protein-protein interaction, lit mining, and curated pathway databases. The resulting STRING score is far more robust than correlation alone. Secondly, the network of 135 proteins surely has a few major functional submodules that can be extracted with tools like by network Markov Clustering (quite easily done in Cytoscape). Finally, would it not be more intuitive to create a single network, layout the major submodules that comprise the 135 DE proteins and color code the node body as the expression level in DCM the node periphery coded by expression in ICM? I guess I find the layout of 3a and 3b to be less optimally informative. I do understand that the networks include metabolites, but nevertheless.

14. Enrichment of programs in females relative to males is addressed. But again this reviewer craves deeper understanding. The authors do not present either a PCA or a factor-ANOVA that might help describe the relative contributions of diagnosis, sex, or race to the overall variance in the data. Sum of squares analysis? F-ratios?. Again, my point with respect to figure 2 apply to 3cd and e. I suspected a multigroup comparison (2- or 3-way ANOVA) organized as a heatmap would be more intuitive.

In closing, though I realize the critique might come across as "you should analyze the data my way", that's not really my intent. Rather, it is meant as a roadmap to address one of the major limitations of the present manuscript – that the bulk of the data is confined to the supplements and not adequately summarized in the manuscript. I do not feel like I have a grip on the extent of the similarities and differences between Donors, ICM, and DCM patients let alone how each is impacted by sex or race or any other aspect of patient history. This is partly due to figure composition and layout and partly due to the abbreviated form of the manuscript that offers little context as to how this dataset distinguishes itself from prior heart failure profiling studies.

Minor Points

This reviewer is confused. I thought iRT peptides were used in the context of MRM or DIA. How are they used in a DDA workflow?

Reviewer #3 (Remarks to the Author):

Specific Comments to Authors

1. Lines 32-34. Actually, there are now a fair number of papers using endomyocardial biopsy to assess functional genomics in the nonfailing and failing human heart, and even more extensive studies in explanted human hearts from prospectively maintained biobanks, with tissue taken at the time of transplantation or organ donation and immediately immersed in liquid N₂ as in this study. Most of these studies dealt with various types of RNA expression rather than proteomics, however. There have been proteomic analyses published from explanted nonfailing and DCM hearts (e.g. Colak et al, PLoS One 2016; not referenced in current manuscript), but this paper represents the largest series and also looks at ischemic cardiomyopathy.

2. Lines 39-41. Perturbed thyroid hormone signaling has been reported before in IDC human LVs.

3. Methods, line 4 "These samples are not post-mortem." These are heart tissue samples removed immediately from hearts explanted at the time of cardiac transplantation or organ donation and placed immediately in liquid N₂. This approach to obtaining and investigating human cardiac tissue samples has been ongoing and published extensively for nearly 40 years, since the early years of cardiac transplantation. "Not post mortem" is a bit misleading since the organ donors are brain dead, and the transplant recipient hearts are headed for the research lab and pathology department. It's true that the cardiac tissue is quite viable in a biologic sense b/c of the preservation methods uses in situ and on explantation, but there is no reason to declare that the methods here are unique or that "Our understanding of heart failure (HF) has been limited by lack of access to pre-mortem human cardiac tissue."

4. I could not find any information on baseline characteristics of the hearts used. At a minimum this would include by-group mean or median age, sex, race, LVEF, NYHA Class, medication classes and storage time. This information could be supplied in a Supplementary table.

We appreciate the opportunity to provide a revised version of our submission NCOMMS-19-28689-T titled "Core Functional Nodes and Sex-Specific Pathways in Human Ischaemic and Dilated Cardiomyopathy".

We sincerely appreciate the Reviewers' helpful suggestions, which we feel have significantly improved the quality of our paper.

We have *substantially* revised our manuscript, addressing *all* of the comments from both Reviewers, which we present in point-by-point format below:

We would ask you to deposit the metabolomics dataset in a community repository such as MetaboLights or Metabolomics Workbench.

Response: We have uploaded all metabolomics data in Metabolomics Workbench, which we refer to in the Methods on Page 21, lines 468-469.

Reviewers' comments:

Reviewer #2 (Remarks to the Author):

Summary

The manuscript describes proteomic and metabolomic datasets from patients with ischemic and dilated cardiomyopathy relative to controls. There is a paucity of high-quality human heart failure proteomic data in the literature, and existing datasets have often been used in furtherance of specific hypotheses, largely relegated to supplements. Human HF proteomes, therefore, remain undercharacterized, particularly as it pertains to etiological and sex differences. Any study that addresses these lucidly would be a welcome advance.

Evaluation Overview

I have little doubt that the authors are sitting on a treasure trove of data. My primary disappointment is that the data is poorly showcased by the manuscript, which appears to be written as a short communique. Notwithstanding the potential impact of the human datasets, the manuscript, as currently laid out and written, would encounter headwinds at a middling proteomics journal, as the data presentation is quite awkward and opaque. This not simply an issue of "cosmetics", as I understand that researchers can reasonably differ on how to present complex data sets. There are also quality control novelty issues. Therefore, I cannot recommend the manuscript for publication in its present form.

Strength: The authors have provided a link that allows the reader to interact with the dataset and observe pathways and networks. The reviewer appreciates the care taken with the layout of the website at <https://mengboli.shinyapps.io/heartomics/>.

Major Issues.

1. Notwithstanding that the journal is called Nature Communications, the choice of reporting the study as a short communique, rather than a regular research article, is problematic. The lay of the research landscape is not adequately framed, the data itself is given short shrift in the figures and the text is highly descriptive rather than quantitative. Essentially the figures themselves are more metadata than actual data (particularly fig 2). The data is confined largely

to the supplements. Finally, posited new findings are not adequately contextualized, given that there are likely hundreds of cardiac profiling studies that have preceded this one.

Response: We thank the reviewer for this suggestion and have expanded the manuscript to a full article so that the background research landscape is appropriately framed, more quantitative data can be taken from the supplemental data to the manuscript, and the findings can be put in context. We would like to re-iterate however, that this extensive profiling study is the first combined proteomic and metabolomic work in advanced failing human left ventricular myocardium with appropriately matched normal controls.

2. The schematic in figure 1, does not do either the data workflow or the patient cohort justice. Firstly, the nature of the patient cohort should be addressed separately in a more detailed table. More on that later. The proteomic and metabolomic workflow is highly generic. The pictogram essentially says “we did proteomics and metabolomics”. It does not say what kind of proteomics (label-free? TMT?) or metabolomics (LC? GC?, a combination?). The green balloon says “Data processing & data analysis”. It does not say what kind of analysis (PCA? Statistical tests? Pathways? Networks?). If the point is to encapsulate the meat of the study in one glance, it could be greatly improved.

Response: We have now included a Table 1 describing patient clinical characteristics. We have updated the schematic that now includes more specific details of the proteomics, metabolomics, and references the statistical analyses performed. We hope the schematic now gives a satisfactory amount of detail.

3. On to patients. Figure 1b and 1c are insufficient documentation of the patient cohort. Ideally, this would be done in two ways. The manuscript would benefit from a table summarizing the patient vital statistics (height, weight, sex, race) but most importantly, documenting the cardiac functional parameters that led to their inclusion. What was the ejection fraction? A supplement containing detailed table documenting each patient, with vital statistics, height, age, weight, sex, ejection fraction, fractional shortening, left ventricle internal diameter, drug regimen, smoking history, etc. would also be useful.

Response: We have now included a table incorporating the key patient clinical and cardiac functional characteristics and an expanded table in the supplement that includes medications and other parameters. The current level of detail included in the table is of equal or greater than the standard in the field (see Goldenberg *et al.*, *Circulation*, 2019, PMID: 30909726 (the Supplemental Table 1 in this publication, shown below)).

4. These major characteristics of the cohort should be incorporated into the text.

Response: The important clinical and cardiac characteristics have been incorporated into the text (Page 5, Lines 112-116).

5. I took a look at the dataset submission. The authors have submitted the raw data files. They have also submitted maxquant archive with all the pertinent peptide and protein particulars. What is lacking (this is common to many submissions) is a simple table of protein quantitation (e.g. sample # vs protein accession), essentially equivalent to a GEO matrix file that any user can open in excel or any other data analysis program. This is key if the dataset is to be mined as a resource to the broader community and not gather dust.

Response: We thank Reviewer 2 for highlighting this. We have now uploaded a table labelled as “Patient Sample Identifications” to match raw files to Sydney Heart Bank identifications. We have also uploaded a separate table labelled as “Protein Quantification Report by DIA” containing protein abundance data for each patient. Please note this required a new submission as we were unable to update the previous and can be found with:

Identifier PXD016617 (username:reviewer17062@ebi.ac.uk; password: DMOONI9I).

6. The proteomic and metabolomic methods could use some fleshing out. As written, it is not clear how the metabolites were identified. MRM yes, but was this against a previously compiled database of standards? a search? What were the instrument settings, tolerances, etc.?

For the proteomics section what were the MS1 and MS2 mass tolerances. It’s not clear what is being quantified. If the authors are using Maxquant, they using iBAQ or LFQ intensities?

Response: We have updated the methods section to provide more detail. For example, we document that the MRM and retention times are based on a library of authentic standards, and have provided the instrument settings in “Supplementary File 1. Instrument Settings Metabolites” in the Supplementary Data. For the proteomics methods, we have provided MS1 and MS2 mass tolerances and all proteomics mass spectrometry information deposited to the ProteomeXchange Consortium via the PRIDE partner repository with the dataset identifier PXD016617 (username: reviewer17062@ebi.ac.uk; password: DMOONI9I). In the proteomics section of Methods we confirm that we used Maxquant and that we used iBAQ and LFQ intensities (Page 22-24, Lines 490-530).

7. Particularly in the proteome, is there any missing data? What was its distribution? How much missing data would prompt exclusion of protein from the analysis? Was missing data imputed? If so how?

Response: We have now updated the methods section to provide these details: missing data and threshold of exclusion, distribution (see below), imputation details (Page 6, Lines 122-123). We did not perform imputation on the proteomic data set.

8. There is no 10,000 ft view or data QC in the form of a principal component analysis (PCA). How noisy are the data sets? How different are the normal, ICM and DCM proteomes and metabolomes. Does diagnosis track with the first principal component?

Response: We thank Reviewer 2 for highlighting this. We have now included a PCA for both proteomics (Fig. 2a) and metabolomics (Fig. 3a), and referenced in the text at (Page 6, Lines 126-129; Page 10, Lines 215-219, respectively) describing separation by first and second principal components of the groups by diagnoses.

9. The authors are to be commended for going with a statistical framework like Limma which increases power. But they have chosen to go with binary comparisons (Limma modified t-test) and volcano plots, when the experimental design screams for a Limma/ANOVA multigroup comparison. This would maximize power by providing more data points for modeling. More importantly, though, it would allow the authors use an intuitive p-value-filtered, hierarchically-clustered heatmap that would showcase clusters defining the similarities and differences between normal, ICM and DCM groups. This would be a substantial improvement over the Venn diagrams in figure 2a, which are more metadata than data. Volcano plots are actual data but don't show the relationships between ICM and DCM. For instance, in the current analysis there are 135 differentially expressed proteins shared between ICM vs. Donor and DCM vs Donor. Are they regulated in the same direction for each etiology? Presumably a mix of up and down. In a Multigroup LIMMA/ANOVA-heatmap layout that would jump out. The same goes for the metabolites.

Response: We thank Reviewer 2 for raising these important points. The reason for binary comparisons was to determine the pathological differences in each disease compared to control. However, we agree with the Reviewer and have used a generalised linear model to perform multigroup comparison (details provided at Page 24, Lines 538-542) and a p-filtered, hierarchical-clustered heatmap, as suggested by the Reviewer, encapsulating all the proteomic (Fig. 2f) and metabolomic (Fig. 3f) data across all three groups (donor, ICM, DCM) and both genders.

We have generated the Volcano plots (Fig. 2b,c; Fig 3b,c) that highlight the significance and fold changes of the results, and also a new plot (Fig. 2e; Fig. 3e) to highlight the fold changes of significantly perturbed proteins and metabolites and whether changes occurred in either ICM or DCM, or both. These new figures are referred to in the Results section starting at Page 6, Lines 129-208 for proteomics results and Page 10, Lines 219-269 for results on the metabolomics data.

10. Figure 2b is a highly uninformative display of metadata, based on the supplements. It shows the overlap between protein and metabolite pathways. Indeed integration of metabolite and protein data is difficult, but there are more sophisticated publically available tools that would likely help the authors substantially. Such tools allow you to load protein and/or gene, and metabolite levels and perform combined pathway enrichment and topological analysis. An example of this would be MetaboAnalyst. Where metabolite and protein data corroborate each other inference is strengthened. If performed downstream of cluster analysis as described above, the insights could be great.

Response: We acknowledge the lack of specificity in Figure 2b. We have generated a new Fig. 4a that provides a *complete* list of pathways regulated at both the protein and metabolite level for ICM and DCM. Furthermore, we used the Pathview R package to perform proteomic and metabolomic integration and visualisation on relevant KEGG pathways. This approach essentially renders the same results as MetaboAnalyst, but is more reproducible and intuitive. This generated an illustrative example of the significant protein and metabolite fold changes in a typical pathway, in this case the Estrogen pathway (Fig. 4b), for both ICM and DCM, and we have added indicators for ICM and DCM. This new figure is referred to in the text in Results (Page 13, Lines 279-286).

11. Or at the very minimum, the authors could find a way to present the tabular data in their supplements graphically in the manuscript, even if it's as simple as a bar graph ranked by $-\log(p\text{-value})$.

Response: We hope that the new figure combining pathway enrichment and topological analysis adequately captures the protein and metabolite changes at the pathway level, as suggested by the reviewer in point 10. Nevertheless, we have also included tables listing the top metabolites and proteins in each disease (Tables 2-5; Pages 6, 9, 11, 12; Lines 124, 188, 228, 245, respectively).

12. Because the authors have eschewed the traditional article format they do not adequately contextualize the significant body of established work on the link between both circulating and local cardiac levels of thyroid hormone and heart failure. This would include (but not be limited to) work suggesting a local cardiac T3 deficiency. E.g. the PNAS paper from several years ago showing that cardiac overexpression of the deiodinase rescues cardiac function while not affecting hypertrophy in a mouse TAC model.

Response: As we have now expanded to a full article format, we have updated the Discussion on local cardiac levels of thyroid hormone and heart failure (Pages 18-19, Lines 399-416), including referencing the paper mentioned by the reviewer (Page 18, Lines 402-403, Page 19, Lines 407-415).

This reviewer is not affiliated in any way. So high cardiac T4 in the authors' study might be consistent with a bottleneck caused by low cardiac de-iodinase activity. It is up to the authors to make the case for the novelty of their finding.

Response: We agree that our finding is consistent with low cardiac deiodinase activity. However, the paper mentioned above is a mouse study demonstrating that over-expressing deiodinase leads to increased T3 levels and can maintain cardiac contractility whilst preventing deterioration of cardiac function and altered gene expression after pressure overload in a TAC model. Whilst interesting, it is quite detached from the clinical scenario, and does not provide direct evidence of decreased deiodinase activity, and altered T4 or T3 levels, in failing *human* cardiac myocardium. The novelty of our data is that we determined elevated human myocardial T4 levels in two different types of heart failure, providing robust evidence in human myocardium of perturbed thyroid hormone in both these types of advanced heart failure.

13. In figure 3. The authors have performed network analysis on correlated proteins and metabolites to find co-regulated nodes. The networks, as presented, are OK, but again there is perhaps a missed opportunity really chunk down the data and present it in a graphically intuitive manner. First, there is nothing wrong with a correlation network, but we now all have access the latest StringdB databases <https://string-db.org/> where the authors could build functional association networks based on gene correlation, protein-protein interaction, lit mining, and curated pathway databases. The resulting STRING score is far more robust than correlation alone. Secondly, the network of 135 proteins surely has a few major functional submodules that can be extracted with tools like by network Markov Clustering (quite easily done in Cytoscape).

Response: We thank Reviewer 2 for making this suggestion. We used the StringdB databases to build functional association networks as suggested by the Reviewer. However, the resultant plot that attempted to over-lay protein-protein interaction on top of correlation data by edge strength, resulted

in an illegible plot where the edges were no longer visible, and node size no longer represented network influence, critical information we did not want to leave out.

As we did not want to lose the correlation data captured by edge thickness and node size, but did want to add more functional information to the association network as suggested by the Reviewer, we have modified our original networks to include functional classification of the nodes. Thereby, the network plots still contain the inherent strengths of a network analysis, with the added benefit of functional node classification. We hope the new network figures contain sufficient information to address the Reviewers comment.

Finally, would it not be more intuitive to create a single network, layout the major submodules that comprise the 135 DE proteins and color code the node body as the expression level in DCM the node periphery coded by expression in ICM? I guess I find the layout of 3a and 3b to be less optimally informative. I do understand that the networks include metabolites, but nevertheless.

Response: We tried to make a single network including ICM and DCM, with node body for DCM and periphery for ICM, but this combined network was overly-complicated and we feel strongly that it would not be legible by the readership.

14. Enrichment of programs in females relative to males is addressed. But again this reviewer craves deeper understanding. The authors do not present either a PCA or a factor-ANOVA that might help describe the relative contributions of diagnosis, sex, or race to the overall variance in the data. Sum of squares analysis? F-ratios?. Again, my point with respect to figure 2 apply to 3cd and e. I suspected a multigroup comparison (2- or 3-way ANOVA) organized as a heatmap would be more intuitive.

Response: As per Response 9 above, we have now included a multigroup comparison and a p-filtered, hierarchical-clustered heatmap encapsulating the differentially expressed proteomic (Fig. 2f) and metabolomic (Fig. 3f) data across all three groups (donor, ICM, DCM) and both genders. This allows direct visualisation of the relative contributions of pathology and gender. Whilst at the global proteomic and metabolomic level we did not see an appreciable effect of gender, when proteins and metabolites were directly assessed for gender interaction effects we did see a range of biologically plausible results, including many X-linked and mitochondrial related protein and metabolite changes. We have kept the direct gender interaction results in Fig. 5c-e.

In closing, though I realize the critique might come across as “you should analyze the data my way”, that’s not really my intent. Rather, it is meant as a roadmap to address one of the major limitations of the present manuscript – that the bulk of the data is confined to the supplements and not adequately summarized in the manuscript. I do not feel like I have a grip on the extent of the similarities and differences between Donors, ICM, and DCM patients let alone how each is impacted by sex or race or any other aspect of patient history. This is partly due to figure composition and layout and partly due to the abbreviated form of the manuscript that offers little context as to how this dataset distinguishes itself from prior heart failure profiling studies.

Response: We do sincerely appreciate the reviewer’s helpful comments to improve clarity and aid interpretation of our manuscript. We hope that the changes we have made adequately reflect the suggestions of Reviewer 2. We have expanded the manuscript to a full article and have also changed the figures according to the above suggestions.

Minor Points

This reviewer is confused. I thought iRT peptides were used in the context of MRM or DIA. How are they used in a DDA workflow?

Response: We apologise for not providing sufficient detail and have updated the methods to indicate the analysis of both a fractionated pooled sample using DDA for spectral library generation and single-shot analysis of each sample using DIA for quantification. The iRT peptides were spiked into both our DDA and DIA runs as it aids retention time alignment to the spectral library.

Reviewer #3 (Remarks to the Author):

Specific Comments to Authors

1. Lines 32-34. Actually, there are now a fair number of papers using endomyocardial biopsy to assess functional genomics in the nonfailing and failing human heart, and even more extensive studies in explanted human hearts from prospectively maintained biobanks, with tissue taken at the time of transplantation or organ donation and immediately immersed in liquid N2 as in this study. Most of these studies dealt with various types of RNA expression rather than proteomics, however. There have been proteomic analyses published from explanted nonfailing and DCM hearts (e.g. Colak et al, PLoS One 2016; not referenced in current manuscript), but this paper represents the largest series and also looks at ischemic cardiomyopathy.

Response: We appreciate Reviewer 3 drawing our attention to these preceding studies, which is now reflected in the Introduction (Page 3, Lines 58-94). We also discuss the specific publication (Colak *et al.*, PLoS One, 2016) mentioned by the Reviewer (Page 3, Lines 72-74).

2. Lines 39-41. Perturbed thyroid hormone signaling has been reported before in IDC human LVs.

Response: We have updated the manuscript to acknowledge this work with the following statements:

“Recent work demonstrated altered expression of two genes in thyroid hormone biogenesis and reduction of T3 in ischaemic myocardium⁴⁰. The same group also demonstrated altered expression of the same genes in DCM myocardium, increased T4, and decreased T3⁴¹. Our work buttresses these findings by directly showing increased T4 in ICM and DCM left ventricular myocardium” (Page 19, Lines 412-416).

3. Methods, line 4 "These samples are not post-mortem." These are heart tissue samples removed immediately from hearts explanted at the time of cardiac transplantation or organ donation and placed immediately in liquid N2. This approach to obtaining and investigating human cardiac tissue samples has been ongoing and published extensively for nearly 40 years, since the early years of cardiac transplantation. "Not post mortem" is a bit misleading since the organ donors are brain dead, and the transplant recipient hearts are headed for the research lab and pathology department. It's true that the cardiac tissue is quite viable in a biologic sense b/c of the preservation methods uses in situ and on explantation, but there is no reason to declare that the methods here are unique or that "Our understanding of heart failure (HF) has been limited by lack of access to pre-mortem human cardiac tissue."

Response: We appreciate the reviewer's comment. By stating that these samples are not post-mortem, we were merely attempting to ensure that this fact is clear, as reviewers have in the past described our samples as post-mortem. The statement that our understanding has been limited by lack of access to

pre-mortem human cardiac tissue was meant to reflect the fact that, for most cardiovascular researchers, access to human left ventricular myocardium is usually not possible. Furthermore, whilst endo-and epicardial biopsies are often sourced for subsequent molecular characterisation, true LV myocardial biopsies are considerably rarer.

Regardless, we have modified our statement to “Restricted access to pre-mortem human left ventricular myocardium is a significant limitation in the study of heart failure (HF)” (Page 3, Lines 63-74).

4. I could not find any information on baseline characteristics of the hearts used. At a minimum this would include by-group mean or median age, sex, race, LVEF, NYHA Class, medication classes and storage time. This information could be supplied in a Supplementary table.

Response: We thank the reviewer for drawing attention to this, and have consequently generated a table (Table 1, Page 5, Lines 112-116) in the manuscript summarizing key clinical and cardiac functional characteristics (gender; age; BMI; ejection fraction; NYHA class; left ventricular end-diastolic dimensions; trans-pulmonary gradient; percentage with hypercholesterolaemia, kidney disease, diabetes, hypertension, and family history of heart failure; and percentage on heart failure medication classes). We would like to highlight that this level of detail exceeds the standard in the field, see below (Goldenberg *et al.*, *Circulation*, 2019, PMID: 30909726 (the Supplemental Table 1 in this publication, shown below)).

Supplemental Table 1

	Control	HF	
		Pre-LVAD	Post-LVAD
N	5	10	10
Age		58.2 ± 9.24	57.7 ± 11.5
Gender		8M/2F	9M/1F
HTN	0	5.50%	5.50%
DM	0	4.40%	5.50%
HLP	0	3.30%	3.30%
ICM	0	60%	60%
DCM	0	40%	40%
EF (%)	>55	17.6 ± 3.5	18.4 ± 4.0

Reviewers' comments:

Reviewer #2 (Remarks to the Author):

The manuscript has been revised and improved substantially over the first version. Notably, inclusion of detailed tables (both patient data and omic), a commitment to detailed methods, and improved figures & analyses are a plus and serve the authors well. The authors have definitely been responsive. Thanks.

The question remains whether the study is really much more informative than any of the large number of heart failure profiling studies that preceded it. The authors do note that this is probably the first profiling study of human ICM and DCM to include both proteomic and metabolomic data, which is nothing to sneeze at.

But has the combined use of proteomics and metabolomics here really moved the needle in our understanding of ICM and DCM? Notwithstanding the substantial improvement in the manuscript, in my opinion, it still falls short in conveying the uniquely novel findings.

In any event, here are some further observations.

1. Inclusion of PCA and heatmaps in fig 2 are an improvement but could still use some cosmetic work to increase readability and bring data into line with commonly accepted practices.

a. For the PCA plot, both the size and pastel colors of symbols make the graph difficult to read clearly. It would be great if the authors could increase symbol size and switch to a bolder color palette to make the symbols pop. Even after substantial magnification, distinguishing powder blue from mauve and rose blush is not easy.

b. It is common practice on PCA plots to include ellipses that define the 95% confidence interval.

c. For the heatmap, the color scale does not adhere to most common norms (red-black-green; yellow-black-blue; red-white-blue). The authors have chosen to use red-yellow-light blue where yellow is around 0. This is problematic since most people interpret yellow as a degree on the red scale, so at first glance it looks as though most proteins are mildly upregulated.

2. The same critique applies to figure 3.

3. Lines 151-152 are a bit cringe-worthy. “In general, the most down-regulated proteins in donor hearts were the most up-regulated in diseased hearts, and vice versa (Fig. 2f)” . If you think about it, this is true for all data in the universe that are presented as relative changes. If what you mean is that there was little divergence between ICM and DCM compared with Normals, you might want to say it that way.

4. I would take issue with aspects and interpretations of the results, discussion, and conclusion. I’ll address each, though perhaps not in order.

The biggest issue with the discussion is that, despite some improvement, the authors still have not done a great job summarizing and referencing the existing HF profiling literature and distinguishing precisely which of their observations are absolutely novel and which are largely confirmatory. Since there are fewer metabolomics studies, I presume most insight would be on on the metabolomic side. Nevertheless, proper contextualization is still lacking all around.

Take line 442-444 of the conclusion.

This is the first study to apply comprehensive “omic” screening to a large cohort of cryopreserved, advanced human ICM and DCM left ventricular myocardium with necessary comparison to appropriately-matched, histopathologically-normal donor hearts.

Though technically true, in the sense that multi-omics including metabolomics has not been done, there is a vast literature of profiling studies using microarrays, NGS and even a few decent proteomics studies. The authors have not cited many of them or really attempted to summarize the pertinent biosignatures or dominant themes from those studies and explicitly lay out the insights that are unique to this study and which observations complement previous ones.

For instance, extracellular matrix remodeling, inflammatory signaling, oxidative stress, mitochondrial dysfunction, and branched chain amino acid metabolism are all widely known and hotly studied.

There is no discussion of convergence/divergence of their data with the recent human proteomic HF data set of Chen et al. Nat Med. 2018 Aug;24(8):1225-1233. doi: 10.1038/s41591-018-0046-2. Epub 2018 Jun 11.

Little mention of other profiling studies of DCM and ICM (of which there so many. i.e. >20 that would be directly pertinent, some dating back about 20 years)

There is no comment on the convergence/divergences between their human study and what has previously been shown from animal models.

No mention of any expression profiling revealing sex differences in HF or are pertinent to the topic

Fermin et al *Circulation: Cardiovascular Genetics*. 2008;1:117–125

Heidecker et al *European Heart Journal*, Volume 31, Issue 10, May 2010, Pages 1188–1196

Kararigas et al *Volume16, Issue11 November 2014 Pages 1160-1167*

Boheler et al. *Volume16, Issue11, November 2014, Pages 1160-1167*

One of the observations the authors deem to be the primary novel one is high T4 levels in the heart suggestive of impaired T3 biosynthesis in the hearts of DCM/ICM patients, yet this has been shown already in human ICM and DCM before, as cited by the authors (Gil-Calyuela et al). In fact, the issue of intra or extra-cardiac thyroid hormone has been a hotbed of research for years, as reviewed by Gerdes, A. M. and G. Iervasi (2010). "Thyroid replacement therapy and heart failure." *Circulation* 122(4): 385-393.

line 447 We revealed numerous novel findings. OK, which ones explicitly have never been seen before?

line 447- network analyses revealed many 448 functional nodes that are informative for future targeted research. Too many to name explicitly?

Reviewer #3 (Remarks to the Author):

My issues have been resolved satisfactorily in the revision

We are interested in the possibility of publishing your study in Nature Communications, but would like to consider your response to these concerns in the form of a revised manuscript before we make a final decision on publication.

We therefore invite you to revise and resubmit your manuscript, taking into account the points raised. Please highlight all changes in the manuscript text file.

Response: We thank the Editor for the opportunity to address these comments and re-submit our manuscript for further consideration.

Reviewers' comments:

Reviewer #2 (Remarks to the Author):

The manuscript has been revised and improved substantially over the first version. Notably, inclusion of detailed tables (both patient data and omic), a commitment to detailed methods, and improved figures & analyses are a plus and serve the authors well. The authors have definitely been responsive. Thanks.

Response: We thank Reviewer #2 for appreciating the significant effort we made addressing the previous comments.

The question remains whether the study is really much more informative than any of the large number of heart failure profiling studies that preceded it. The authors do note that this is probably the first profiling study of human ICM and DCM to include both proteomic and metabolomic data, which is nothing to sneeze at.

Response: Indeed, as described in more detail below, ours is the first study to comprehensively assess human left ventricular myocardium with appropriately matched (age, gender, and BMI) controls, in large sample numbers, in two different heart failure aetiologies, at both the protein and metabolite level. As we also discuss in further detail below, one of the advantages of using both the protein and metabolite data layers is that they are more “adjacent” to phenotype, and as such may generate more translatable results than studies based on only, say, transcriptomic data, where many of the changes are not propagated *via* translation to the protein or further downstream to the metabolite. Furthermore, due to our matched cases vs control study design, we were able to uncover novel interactions of gender with heart failure, including X-linked metabolites such as ornithine, mitochondrial substrates such as methylbutyrylcarnitine, and mediators of nitric oxide synthase (NOS) such as ADMA, the major endogenous inhibitor of NOS. Finally, we created an interactive online application that serves to enable the research efforts of the broader cardiovascular community, the vast majority of whom do not have access to *human* left ventricular myocardial tissue.

But has the combined use of proteomics and metabolomics here really moved the needle in our understanding of ICM and DCM? Notwithstanding the substantial improvement in the manuscript, in my opinion, it still falls short in conveying the uniquely novel findings.

In any event, here are some further observations.

1. Inclusion of PCA and heatmaps in fig 2 are an improvement but could still use some cosmetic work to increase readability and bring data into line with commonly accepted practices.

a. For the PCA plot, both the size and pastel colors of symbols make the graph difficult to read clearly. It would be great if the authors could increase symbol size and switch to a bolder color palate to make the symbols pop. Even after substantial magnification, distinguishing powder blue from mauve and rose blush is not easy.

Response: We thank the Reviewer for this helpful suggestion, and have increased symbol size and changed to a bolder colour scheme accordingly.

b. It is common practice on PCA plots to include ellipses that define the 95% confidence interval.

Response: We apologize for this omission and have added ellipses to define the 95% confidence interval.

c. For the heatmap, the color scale does not adhere to most common norms (red-black-green; yellow-black-blue; red-white-blue). The authors have chosen to use red-yellow-light blue where yellow is around 0. This is problematic since most people interpret yellow as a degree on the red scale, so at first glance it looks as though most proteins are mildly upregulated.

2. The same critique applies to figure 3.

Response: We thank the Reviewer for this excellent suggestion and have updated the colour scheme for the heatmaps (Fig. 2f, Fig. 3f) accordingly, with white being the new zero-point.

3. Lines 151-152 are a bit cringe-worthy. “In general, the most down-regulated proteins in donor hearts were the most up-regulated in diseased hearts, and vice versa (Fig. 2f)” . If you think about it, this is true for all data in the universe that are presented as relative changes. If what you mean is that there was little divergence between ICM and DCM compared with Normals, you might want to say it that way.

Response: We agree, this was not the best way to describe these changes. Indeed, we have already stated that the majority of proteins were regulated in both ICM and DCM relative to donor hearts. We have therefore decided to delete this sentence.

4. I would take issue with aspects and interpretations of the results, discussion, and conclusion. I'll address each, though perhaps not in order.

The biggest issue with the discussion is that, despite some improvement, the authors still have not done a great job summarizing and referencing the existing HF profiling literature and distinguishing precisely which of their observations are absolutely novel

and which are largely confirmatory. Since there are fewer metabolomics studies, I presume most insight would be on the metabolomic side. Nevertheless, proper contextualization is still lacking all around.

Response: We thank the Reviewer for highlighting this important point, which we have devoted considerable effort to address in this revision. We have added text to the discussion to summarize more thoroughly the existing HF profiling literature and clearly stating which of our results are novel and which are confirmatory. In order to ensure we have included all relevant studies, we have made the table below that compiles the key features of all previous proteomic and metabolomic HF profiling studies and included our own as a comparator. This table highlights that our study is the largest integrated proteomic and metabolomic study ever performed that includes age-matched controls of both genders from pre-mortem tissue. Our paper is unique as it appropriately matched cases with controls across critical confounders, integrates both data layers using correlation network analysis, interrogates the interaction of gender with heart failure, and provides a web-based resource available to the scientific community.

Summary of human heart failure proteomic and metabolomic profiling studies

Author/Year	Hearts, n	Aetiology	Age, Gender, BMI Matched Controls ?	Gender	Examined Gender interaction?	Web-based Resource?	Data Layer
This paper	44. 15 ICM, 14 DCM, 15 Control.	ICM, DCM	Yes	DCM & ICM 66% Male; Donor 50% Male.	Yes	Yes	Protein, Metabolite
Du et al., 2019 ¹	14. 7 LVNC, 7 ARVC	LVNC, ARVC	No normal controls.	79% Male	No	No	Protein
Coats et al., 2018 ²	17. 11 HCM, 6 Controls.	HCM.	No	HCM 27% Male; Controls 84% Male.	No	No	Protein
Chen et al., 2018 ³	33. 7 Control, 6 cHyp, 6 ICM, 4 HCMpEF, 5 HCMrEF, 6 DCM.	LV hypertrophy, HCM, DCM, ICM.	No	Controls 84% male, cHyp 50% male, ICM 67%, HCMpEF 100% male, HCMrEF 60% male, DCM 67% male.	No	No	Protein
Chen et al., 2017 ⁴	14. Protein: RV and LV from 4ARVC, 4 DCM, 4 controls. RNA: RV from 10 ARVC, 4	ARVC.	No	ARVC 80% Male, DCM 60% Male, Controls, gender not provided.	No	No	Protein, RNA

	controls.						
Ameling et al., 2016 ⁵	33 DCM (RNA), 23 DCM (protein)	DCM. Pre/post immunoadsorption	No normal controls.	78% Male.	No	No	RNA, protein
Oda et al., 2015 ⁶	16 Right atrium samples from 16 patients pre, during, and post hypo/normothermic cooling at CBP.	Aortic aneurysm	No normal controls.	54% Male.	No	No	Protein
Menazza et al., 2015 ⁷	LV from 11 DCM and 6 control	DCM	No	DCM 64% Male; 33% Male.	No	No	S-nitrosylated proteins.
Zitta et al., 2015 ⁸	Right atrial tissue from 19 patients with, and 19 without, RIPC before CPB.	CBP for CABG or AVR.	No	Control 100% Male; RIPC 78% Male.	No	No	Protein
Schechter et al., 2014 ⁹	LV from explanted hearts: 4 non-failing; 4 ischaemic failing; 4 non-ischaemic failing	ICM, non-ICM	No	Control 100% Male; ICM 100% Male; Non-ICM 100% Male	No	No	Phosphoproteome
Helms et al., 2014 ¹⁰	LV HCM (n=46) myocardium at myomectomy, LVAD implantation, or explanted hearts, or unmatched donor hearts (n=10)	HCM	No	Control 40% Male; HCM 50-64% Male	No	No	RNA, Protein
Zheng et al., 2014 ¹¹	9 RHD and 9 control (with prolapse) MV papillary muscle.	Rheumatic heart disease	Age and gender matched	Control 30% Male, RHD 30% Male	No	No	Protein
Lionetti et al., 2014 ¹²	LV myocardium from 11 DCM, 12 ICM, 8 Donor	ICM, DCM	No	Donor 82% Male; DCM 91% Male; ICM 100% Male	No	No	Protein
Kakimoto et al., 2013	Post-mortem LV in 5 AMI and Controls	Post-mortem LV post acute myocardial infarction	Age and gender matched	Control 80% Male, AMI 80% Male	No	No	Protein
Brioschi et al., 2012 ¹³	LV from 14 Failing	ICM, DCM	No	Failing 43% Male,	No	No	Carbonylated Proteins

	(7DCM, 7 ICM) and 13 non-Failing hearts			Control 54% Male			
Zhang et al., 2011 ¹⁴	22 Post-mortem LV myocardium; 14 explanted LV myocardium	Hypertrophy, Severe hypertrophy/dilatation, CHF	No	Control: 43% Male, Hypertrophy 40% Male, Severe hypertrophy/dilatation 75% Male, CHF 67%	No	No	Protein
Hammer et al., 2011 ¹⁵	10 iDCM and 7 control endomyocardial biopsies	Inflammatory Dilated Cardiomyopathy	No	iDCM 60% Male; Control 71% Male	No	No	Protein
Urbonavicius et al., 2009 ¹⁶	Endomyocardial biopsies from 9 CHF, 4 endstage failing myocardium	Reversible vs irreversible dysfunctional myocardium	No	NA	No	No	Protein
Mayr et al., 2008 ¹⁷	Atrial appendage from 8 patients who maintained sinus rhythm and 7 who developed AF post valve surgery	Atrial myocardial metabolic demand of AF	No	Sinus rhythm 88% Male, AF 88% Male	No	No	Protein, Metabolite
Teixeira et al., 2006 ¹⁸	LV myocardium from 2 CCC patient	Chronic Chagas disease cardiomyopathy	No	100% Female	No	No	Protein

LV: left ventricle; RV: right ventricle; LVNC: left-ventricular non-compaction; ARVC: arrhythmogenic right ventricular cardiomyopathy; HCM: hypertrophic cardiomyopathy; DCM: dilated cardiomyopathy; ICM: ischaemic cardiomyopathy; cHyp: compensated hypertrophy; HCMpEF: hypertrophic cardiomyopathy with preserved ejection fraction; HCMrEF: hypertrophic cardiomyopathy with reduced ejection fraction; CBP: cardiopulmonary bypass; RIPC: remote ischaemic preconditioning; MV: mitral valve; RHD: rheumatic heart disease; CHF: congestive heart failure; iDCM: inflammatory dilated cardiomyopathy; AF: atrial fibrillation; CCC: Chronic Chagas' disease cardiomyopathy.

Furthermore, the following additional text has been added to the Discussion, to expand on our findings that validate those reported before:

Regarding our findings of changes in extracellular matrix proteins, we have contextualized our findings as follows: “*We confirmed established pathogenesis in ICM and DCM including changes in many ECM proteins, a well-established feature of cardiac remodelling¹⁹, and in ICM changes in von Willebrand factor and serotonin indicating platelet activation, along*

with components of the blood coagulation cascade, highlighting the atherothrombotic pathogenesis of ICM.” (Page 16, Lines 337-340).

We have added the following to place the novelty of the SAA1 result in context:

“In ICM hearts, serum amyloid A was decreased 10-fold, the first report of this protein perturbation in human ICM myocardium.” (Page 16, Lines 340-341). “Supportive of this finding, Chen et al. recently determined that SAA1 gene expression was significantly reduced in ICM myocardium and SAA1 protein differentially expressed between DCM, ICM, and hypertrophic cardiomyopathy⁴.” (Page 16, Lines 345-347).

We have added the following text to discuss our thyroid hormone result more thoroughly:

“It has also been suggested that the changes in local cardiac conversion of T4 to T3 are less relevant than the reduced plasma T3 levels in heart failure, as the majority of cardiac T3 comes from the circulation, with only up to 7% resulting from conversion of T4 by D2 deiodinase in the heart²⁰. However, our data is supportive of the Gil-Cayuela et al. data^{21, 22}, and together these data suggest that in heart failure myocardium, low cardiac T3 coming from the circulation is likely exacerbated by reduced local conversion by less active D2 deiodinase, which led to the elevated T4 we saw in the myocardium.” (Page 17, Lines 363-370).

The following text has been added in the Discussion in cases where our findings are truly novel:

“Whilst decreased FMN has been reported in canine myocardium post ischaemia²³, this is the first report to our knowledge of changes in this cofactor in human heart failure myocardium. Furthermore, it has also been demonstrated in rat hearts that administration of FMN, along with other riboflavin derivatives, is protective post ischaemia-reperfusion via marked recovery of high energy phosphate compounds and pH in the extracellular space²⁴. Administration of related metabolite cytoflavin was shown to be cardioprotective in doxorubicin-induced cardiomyopathy in rats²⁵. Therefore, there is pre-clinical evidence that FMN is reduced in ischaemic myocardium, and that administration of FMN and / or related metabolites is cardioprotective in cardiac ischaemia and toxic cardiomyopathy²³⁻²⁶. However, we are the first to show a decrease in FMN in human heart failure myocardium. In fact, FMN had the greatest fold change and significance of all metabolites in both ICM and DCM myocardium; whether it is a viable therapeutic target in the treatment of these two most common human heart failure aetiologies remains to be seen.” (Pages 17-18, Lines 376-388)

Further contextualising the purine and pyrimidine results, and highlighting the novelty, we have added the following:

“Here we show these changes in the myocardium itself, confirming that these peripheral changes in the circulation of heart failure patients were most likely derived from perturbations of these substrates in the heart.” (Page 18, Lines 392-394)

Explicitly highlighting novelty of our gender-heart failure interaction results, we have added the following:

“Recent work has highlighted the differences in gene expression between male and female hearts²⁷, with several exploring the interaction of gender with heart failure²⁸⁻³¹. Using left ventricular myocardium derived from explanted hearts or obtained at the time of left ventricular assist device insertion, Fermin et al. explored cardiac sexual dimorphism at the gene transcript level and reported >1800 genes showing sexual dimorphism in the heart, many involved in ion transport or G-coupled receptor signaling²⁸. Using ventricular myocardium from 21 failing compared to 4 non-failing hearts at transplant, Boheler et al. found significant interactions between heart failure and gender, and transcript changes that were specific to gender, e.g. *Gpd58* decreased in male HF but increased in female HF²⁹. Kararigas et al. found 36 17 β -estradiol E2-dependent genes regulated in a sex-specific manner³⁰, after exposing epi-myocardium obtained at the time of coronary artery bypass grafting to E2 or hydroxypropyl- β -cyclodextrin. In both ICM and DCM myocardium, Y-chromosome related protein *EIF1AY* was dramatically elevated in male vs female heart failure myocardium, having previously been reported at the transcript level only³¹. In our study, we saw several novel gender interactions at both the protein and metabolite level. We determined several mitochondrial-related gender interactions: e.g. *FMN*, which is a co-factor in mitochondrial oxidoreductase reactions, and mitochondrial substrates like short-chain acylcarnitines, all consistent with the gender-specificity of mitochondrial function^{32, 33}. Notably, inherited short-chain acylcarnitine disorders, e.g. carnitine palmitoyl transferase II deficiency, are inherited in an autosomal recessive manner, but predominantly affect males. Ornithine, metabolized by X chromosome-encoded ornithine transcarbamylase, had a significant gender interaction. A divergent effect of androgens on ornithine decarboxylase activity in rat hearts has been reported before³⁴, but to our knowledge we are the first to report ornithine’s interaction with gender in human heart failure myocardium. TMAO was significantly elevated in male heart failure myocardium only, consistent with its metabolism by gender-specific flavin-containing monooxygenases³⁵. We are the first to report the interaction of TMAO with gender in human heart failure myocardium, which has important implications when considering that this microbiome-derived metabolite has been reported widely as a mediator of atherosclerosis, platelet activation, and myocardial infarction³⁵⁻³⁹; our data suggests it may be a contributor to the divergence in heart failure outcomes in men and women. An interaction of nitric oxide deficiency with gender in recovery post ischaemia has been reported before in mice⁴⁰, and S-nitrosoglutathione reductase was reported to be a critical sex-dependent mediator of myocardial protein S-nitrosylation in mice⁴¹ and murine ex vivo hearts⁴² (more active in females); S-nitrosylation is reported as an essential mediator of nitric oxide-dependent cardioprotection⁴¹. Other work in mice revealed that nitroglycerin-induced calcitonin gene-related peptide release is eNOS-dependent, with a greater response in females⁴³, and estrogen has been shown to upregulate NOS in neonatal rat cardiomyocytes⁴⁴. However, we are the first to report that in both ICM and DCM, inhibitors of nitric oxide synthase (*ADMA* and *L-NMMA*, respectively) were increased in male hearts only, implicating perturbed NOS activity as a predominant driver of male, but not female, HF. Therefore, we report novel insights into the interaction of gender with heart failure, and extended observations only seen previously at the transcript level or in animal studies.”

(Pages 19-20, Lines 412-453; Page 15, Lines 324-328)

Take line 442-444 of the conclusion.

This is the first study to apply comprehensive “omic” screening to a large cohort of

cryopreserved, advanced human ICM and DCM left ventricular myocardium with necessary comparison to appropriately-matched, histopathologically-normal donor hearts.

Though technically true, in the sense that multi-omics including metabolomics has not been done, there is a vast literature of profiling studies using microarrays, NGS and even a few decent proteomics studies. The authors have not cited many of them or really attempted to summarize the pertinent biosignatures or dominant themes from those studies and explicitly lay out the insights that are unique to this study and which observations complement previous ones.

Response: We hope the additional text in the manuscript, outlined in the above response, has adequately summarized the pertinent biosignatures or dominant themes from those studies, and made clear which of our insights are unique and which complement previous work.

In addition, at the end of the Introduction we have highlighted the signatures coming from previous work, and described the purpose of our study to advance the field, as follows:

“Studies examining the functional genomic signature of DCM and ICM explanted hearts have generally concentrated on “upstream” domains⁴⁵, incorporating whole-genome sequencing, transcriptomic profiling, and sometimes protein expression. This work has provided novel insights into molecular changes in human heart failure, implicating extracellular matrix remodelling, inflammatory signalling, oxidative stress, mitochondrial dysfunction, and branched chain amino acid metabolism as signature pathogenic changes^{3, 7, 46}. The proteomic studies usually used small numbers, infrequently more than one aetiology, and without matching across age, gender, and BMI^{1-18, 47}. Further, a more extensive understanding as to whether upstream perturbations are propagated downstream via translation to the protein level, and further via enzymatic processing to the metabolite level, are required. Such analyses require large sample sizes, in addition to matched control (non-diseased) groups to adequately address the major confounders of age, gender, and body-mass index (BMI). Therefore, we determined the perturbations in ICM and DCM at the protein and metabolite pathway level using an unbiased and comprehensive screen of a large number of heart failure samples matched to histopathologically-normal donor controls for age, gender, and BMI – a total of 51 left ventricular myocardial samples from 44 human hearts (Table 1; Fig. 1a-c). Quantification of proteins and metabolites, both downstream of genetic variation, captures the contribution of genes, environment, and their interaction, and serve as a rich source of “translatable” diagnostic and therapeutic targets. Furthermore, multi-omic integration reveals interplay between different layers of a biological system such as metabolites with enzymes/transporters⁴⁸, and can discover new associations between these biological layers. We provide all proteomic and metabolomic results via an interactive online repository (<https://mengboli.shinyapps.io/heartomics/>) as a publicly available resource, thereby enabling researchers without access to human cardiac tissue.” (Page 3-4, Lines 73-96)

For instance, extracellular matrix remodeling, inflammatory signaling, oxidative stress, mitochondrial dysfunction, and branched chain amino acid metabolism are all widely known and hotly studied.

Response: As above, we hope that our additions to the Discussion now adequately reference the previous work in this area.

There is no discussion of convergence/divergence of their data with the recent human proteomic HF data set of Chen et al. Nat Med. 2018 Aug;24(8):1225-1233. doi: 10.1038/s41591-018-0046-2. Epub 2018 Jun 11.

Response: We have now referenced and contextualized this paper in our Discussion. Whilst this was a very thorough study, there were only 7 normal (non-hypertrophied), 6 ICM, and 6 DCM hearts, and the BMIs were dramatically different between groups (e.g. mean BMI of DCMs was 36, whilst that of ICM was 27 and donors 28.5). There were several interesting points of convergence of our data with this paper. For example, in both the Chen *et al.* paper and ours, APOA4 was significantly upregulated in DCM myocardium, at the gene expression level in their paper, and at the protein level in ours. Furthermore, both the Chen *et al.* and our paper had significant increases in PRELP and MFAP4 in ICM hearts, and members of the NDUF, TF, and COL families of proteins in ICM and DCM hearts. Intriguingly, the Chen *et al.* data included changes in SAA1 in ICM hearts at the gene expression level, which was the most significantly changed protein in our ICM hearts (discussed above). However, the Chen *et al.* paper did not report on thyroid hormone pathway changes: this is likely because our metabolite data captured the myocardial increase in T4 at the metabolite level; the lack of proteomic changes in both our and the Chen *et al.* data likely represents the difficulties measuring the deiodinase proteins in this pathway using standard proteomic extraction and analytic techniques. We have added the following text to the Discussion:

“Our analysis also informs observations made previously at the transcript and protein level. For example, recent work by Chen et al.³ found APOA4 mRNA to be upregulated in DCM hearts. We importantly confirm that this regulation of APOA4 is maintained at the protein level. Furthermore, our data validates other protein expression changes in this paper, such as the upregulation of proteins PRELP and MFAP4 in ICM myocardium, along with members of the NDUF, TF, and COL families of proteins in both ICM and DCM myocardium³. There were also many differences with the Chen et al. study³, for example the thyroid hormone pathway; this is likely because our metabolite data captured the myocardial increase in T4; the lack of proteomic changes in both ours and the Chen et al. dataset likely represents the difficulties measuring deiodinase proteins using standard proteomic extraction and analytic techniques.” (Pages 18-19, Lines 401-411).

Little mention of other profiling studies of DCM and ICM (of which there so many. i.e. >20 that would be directly pertinent, some dating back about 20 years)

Response: We hope that our additions to the Discussion, as listed above, now adequately capture the other profiling studies of DCM and ICM.

There is no comment on the convergence/divergences between their human study and what has previously been shown from animal models.

Response: We have deliberately concentrated on previous human studies, as they are more directly relevant. Furthermore, species difference in certain pathways are critically different, for example, regarding the thyroid hormone pathway, D2 deiodinase is not expressed in

rodents, the critical enzyme responsible for local cardiac hypothyroidism in heart failure.

No mention of any expression profiling revealing sex differences in HF or are pertinent to the topic

Fermin et al *Circulation: Cardiovascular Genetics*. 2008;1:117–125

Heidecker et al *European Heart Journal*, Volume 31, Issue 10, May 2010, Pages 1188–1196

Kararigas et al *Volume16, Issue11 November 2014 Pages 1160-1167*

Boheler et al. *Volume16, Issue11, November 2014, Pages 1160-1167*

Response: We thank the Reviewer for highlighting this omission of previous important work in this area, and have extended our discussion, including each of the above-mentioned references. The following text has been added to the Discussion:

*“Recent work has highlighted the differences in gene expression between male and female hearts²⁷, with several exploring the interaction of gender with heart failure²⁸⁻³¹. Using left ventricular myocardium derived from explanted hearts or obtained at the time of left ventricular assist device insertion, Fermin et al. explored cardiac sexual dimorphism at the gene transcript level and reported >1800 genes showing sexual dimorphism in the heart, many involved in ion transport or G-coupled receptor signaling²⁸. Using ventricular myocardium from 21 failing compared to 4 non-failing hearts at transplant, Boheler et al. found significant interactions between heart failure and gender, and transcript changes that were specific to gender, e.g. *Gpd58* decreased in male HF but increased in female HF²⁹. Kararigas et al. found 36 17 β -estradiol E2-dependent genes regulated in a sex-specific manner³⁰, after exposing epi-myocardium obtained at the time of coronary artery bypass grafting to E2 or hydroxypropyl- β -cyclodextrin. In both ICM and DCM myocardium, Y-chromosome related protein *EIF1AY* was dramatically elevated in male vs female heart failure myocardium, having previously been reported at the transcript level only³¹.” (Page 19, Lines 412-425).*

One of the observations the authors deem to be the primary novel one is high T4 levels in the heart suggestive of impaired T3 biosynthesis in the hearts of DCM/ICM patients, yet this has been shown already in human ICM and DCM before, as cited by the authors (Gil-Calyuela et al). In fact, the issue of intra or extra-cardiac thyroid hormone has been a hotbed of research for years, as reviewed by Gerdes, A. M. and G. Iervasi (2010). "Thyroid replacement therapy and heart failure." *Circulation* 122(4): 385-393.

Response: We agree with the Reviewer that this observation has been made before. However, we did not explicitly state that this is a new finding. We did highlight this result in the abstract as we felt it was an important finding to highlight as it occurred in both pathologies. Furthermore, in the discussion we gave primacy to the above referenced publication, and said our result validates these findings. Our work, therefore, validates these results and reveals comparable elevation in myocardial T4 levels in both ICM and DCM hearts in the same study.

In the above review paper by Gerdes in *Circulation*, it is stated that in low-T3 syndrome, there is a pronounced impairment of cardiac function, and low plasma T3 is prognostically significant for heart failure outcomes. This also suggests that *peripheral* conversion of T4 to

T3 is reduced, and that less T3 is available for the heart. Our data, and that of Gil-Calvuela *et al.*, suggests that local cardiac D2 deiodinase activity also contributes, as this results in the increased T4/T3 ratio seen in our study and theirs. Indeed, it has been shown, that cardiac D2 is highly responsive to low circulating T3 levels in normal hearts (Wagner *et al.*, *J Mol Endocrinol*, 2003), but this may be compromised in failing myocardium.

We have added the following text to the Discussion:

*“It has also been suggested that the changes in local cardiac conversion of T4 to T3 are less relevant than the reduced plasma T3 levels in heart failure, as the majority of cardiac T3 comes from the circulation, with only up to 7% resulting from conversion of T4 by D2 deiodinase in the heart²⁰. However, our data is supportive of the Gil-Calvuela *et al.* data^{21, 22}, and together these data suggest that in heart failure myocardium, low cardiac T3 coming from the circulation is likely exacerbated by reduced local conversion by less active D2 deiodinase, which led to the elevated T4 we saw in the myocardium.” (Page 17, Lines 363-370).*

line 447 We revealed numerous novel findings. OK, which ones explicitly have never been seen before?

Response: We hope that the additional text in the Discussion, and outlined above in the Response, sufficiently addresses the Reviewer’s query regarding which findings explicitly are truly novel.

In the Conclusion, we have added the following text:

“We report perturbations in well-established pathological pathways in heart failure myocardium such as fibrotic remodeling and oxidative stress, and confirm previous perturbations in local cardiac thyroid metabolism. Many of the perturbed pathways were common to both heart failure aetiologies, revealing overlapping molecular changes as heart failure advances. We report novel changes in ICM and DCM myocardium, including serum amyloid A1 protein, FMN, purines and pyrimidines, and novel gender-specific perturbations in histidine, acylcarnitines, ornithine, microbiome-derived atherogenic factor TMAO, and nitric oxide metabolites, permitting a better understanding of divergent HF pathogenesis between males and females.” (Pages 20-21, Lines 457-465)

line 447- network analyses revealed many 448 functional nodes that are informative for future targeted research. Too many to name explicitly?

Response: To visualise these functional nodes and improve accessibility to readers, we have created a web-based resource. Furthermore, in Fig 5 we have also clustered the nodes into their DAVID annotated pathways and explicitly highlighted nodes annotated to complement and coagulation (light green/blue), extracellular matrix-associated (purple), cytoskeletal (orange), and immune (yellow). The size of the nodes represents network influence, so for example in ICM a cluster of ECM proteins such as EFEMP1, LUM, and COL6A2 are highly influential, as is TF in the coagulation cascade.

We have added the following text to the Conclusion:

“Our networks succinctly illustrate the most influential nodes in both types of heart failure; for example, in ICM myocardium, a cluster of ECM proteins such as EFEMP1, LUM, and COL6A2 are highly influential, as is TF in the coagulation cascade.” Page 21, Lines 465-468)

Reviewer #3 (Remarks to the Author):

My issues have been resolved satisfactorily in the revision

Response: We are happy that we have satisfactorily addressed this Reviewer’s comments.

REFERENCES:

1. Du H, Liu S, Li C and Wei Y. Comparative proteomics analysis of myocardium in left ventricular non-compaction cardiomyopathy. *Acta Biochim Biophys Sin (Shanghai)*. 2019;51:653-655.
2. Coats CJ, Heywood WE, Virasami A, Ashrafi N, Syrris P, Dos Remedios C, Treibel TA, Moon JC, Lopes LR, McGregor CGA, Ashworth M, Sebire NJ, McKenna WJ, Mills K and Elliott PM. Proteomic Analysis of the Myocardium in Hypertrophic Obstructive Cardiomyopathy. *Circ Genom Precis Med*. 2018;11:e001974.
3. Chen CY, Caporizzo MA, Bedi K, Vite A, Bogush AI, Robison P, Heffler JG, Salomon AK, Kelly NA, Babu A, Morley MP, Margulies KB and Prosser BL. Suppression of deetyrosinated microtubules improves cardiomyocyte function in human heart failure. *Nat Med*. 2018;24:1225-1233.
4. Chen L, Yang F, Chen X, Rao M, Zhang NN, Chen K, Deng H, Song JP and Hu SS. Comprehensive Myocardial Proteogenomics Profiling Reveals C/EBPalpha as the Key Factor in the Lipid Storage of ARVC. *J Proteome Res*. 2017;16:2863-2876.
5. Ameling S, Bhardwaj G, Hammer E, Beug D, Steil L, Reinke Y, Weitmann K, Grube M, Trimpert C, Klingel K, Kandolf R, Hoffmann W, Nauck M, Dorr M, Empen K, Felix SB and Volker U. Changes of myocardial gene expression and protein composition in patients with dilated cardiomyopathy after immunoadsorption with subsequent immunoglobulin substitution. *Basic Res Cardiol*. 2016;111:53.
6. Oda T, Yamaguchi A, Shimizu K, Nikai T and Matsumoto K. Does the Rewarmed Heart Restore the Myocardial Proteome to That of the Pre-Cooled State?--A Proteomic Analysis of Surgical Samples. *Circ J*. 2015;79:2648-58.
7. Menazza S, Aponte A, Sun J, Gucek M, Steenbergen C and Murphy E. Molecular Signature of Nitroso-Redox Balance in Idiopathic Dilated Cardiomyopathies. *J Am Heart Assoc*. 2015;4:e002251.
8. Zitta K, Meybohm P, Gruenewald M, Cremer J, Zacharowski KD, Scholz J, Steinfath M and Albrecht M. Profiling of cell stress protein expression in cardiac tissue of cardiac surgical patients undergoing remote ischemic preconditioning: implications for thioredoxin in cardioprotection. *J Transl Med*. 2015;13:34.
9. Schechter MA, Hsieh MK, Njoroge LW, Thompson JW, Soderblom EJ, Feger BJ, Troupes CD, Hershberger KA, Ilkayeva OR, Nagel WL, Landinez GP, Shah KM, Burns VA, Santacruz L, Hirschey MD, Foster MW, Milano CA, Moseley MA, Piacentino V, 3rd and Bowles DE. Phosphoproteomic profiling of human myocardial tissues distinguishes ischemic from non-ischemic end stage heart failure. *PLoS One*. 2014;9:e104157.
10. Helms AS, Davis FM, Coleman D, Bartolone SN, Glazier AA, Pagani F, Yob JM, Sadayappan S, Pedersen E, Lyons R, Westfall MV, Jones R, Russell MW and Day SM. Sarcomere mutation-specific expression patterns in human hypertrophic cardiomyopathy. *Circ Cardiovasc Genet*. 2014;7:434-43.
11. Zheng D, Xu L, Sun L, Feng Q, Wang Z, Shao G and Ni Y. Comparison of the ventricle muscle proteome between patients with rheumatic heart disease and controls with mitral valve prolapse: HSP 60 may be a specific protein in RHD. *Biomed Res Int*. 2014;2014:151726.

12. Lionetti V, Matteucci M, Ribezzo M, Di Silvestre D, Brambilla F, Agostini S, Mauri P, Padeletti L, Pingitore A, Delsedime L, Rinaldi M, Recchia FA and Pucci A. Regional mapping of myocardial hibernation phenotype in idiopathic end-stage dilated cardiomyopathy. *J Cell Mol Med*. 2014;18:396-414.
13. Brioschi M, Polvani G, Fratto P, Parolari A, Agostoni P, Tremoli E and Banfi C. Redox proteomics identification of oxidatively modified myocardial proteins in human heart failure: implications for protein function. *PLoS One*. 2012;7:e35841.
14. Zhang J, Guy MJ, Norman HS, Chen YC, Xu Q, Dong X, Guner H, Wang S, Kohmoto T, Young KH, Moss RL and Ge Y. Top-down quantitative proteomics identified phosphorylation of cardiac troponin I as a candidate biomarker for chronic heart failure. *J Proteome Res*. 2011;10:4054-65.
15. Hammer E, Goritzka M, Ameling S, Darm K, Steil L, Klingel K, Trimpert C, Herda LR, Dorr M, Kroemer HK, Kandolf R, Staudt A, Felix SB and Volker U. Characterization of the human myocardial proteome in inflammatory dilated cardiomyopathy by label-free quantitative shotgun proteomics of heart biopsies. *J Proteome Res*. 2011;10:2161-71.
16. Urbonavicius S, Wiggers H, Botker HE, Nielsen TT, Kimose HH, Ostergaard M, Lindholt JS, Vorum H and Honore B. Proteomic analysis identifies mitochondrial metabolic enzymes as major discriminators between different stages of the failing human myocardium. *Acta Cardiol*. 2009;64:511-22.
17. Mayr M, Yusuf S, Weir G, Chung YL, Mayr U, Yin X, Ladroue C, Madhu B, Roberts N, De Souza A, Fredericks S, Stubbs M, Griffiths JR, Jahangiri M, Xu Q and Camm AJ. Combined metabolomic and proteomic analysis of human atrial fibrillation. *J Am Coll Cardiol*. 2008;51:585-94.
18. Teixeira PC, Iwai LK, Kuramoto AC, Honorato R, Fiorelli A, Stolf N, Kalil J and Cunha-Neto E. Proteomic inventory of myocardial proteins from patients with chronic Chagas' cardiomyopathy. *Braz J Med Biol Res*. 2006;39:1549-62.
19. Kim GH, Uriel N and Burkhoff D. Reverse remodelling and myocardial recovery in heart failure. *Nat Rev Cardiol*. 2018;15:83-96.
20. Gerdes AM and Iervasi G. Thyroid replacement therapy and heart failure. *Circulation*. 2010;122:385-93.
21. Gil-Cayueta C, Rosello LE, Tarazon E, Ortega A, Sandoval J, Martinez-Dolz L, Cinca J, Jorge E, Gonzalez-Juanatey JR, Lago F, Rivera M and Portoles M. Thyroid hormone biosynthesis machinery is altered in the ischemic myocardium: An epigenomic study. *Int J Cardiol*. 2017;243:27-33.
22. Gil-Cayueta C, Ortega A, Tarazon E, Martinez-Dolz L, Cinca J, Gonzalez-Juanatey JR, Lago F, Rosello-Lleti E, Rivera M and Portoles M. Myocardium of patients with dilated cardiomyopathy presents altered expression of genes involved in thyroid hormone biosynthesis. *PLoS One*. 2018;13:e0190987.
23. Rouslin W and Ranganathan S. Impaired function of mitochondrial electron transfer complex I in canine myocardial ischemia: loss of flavin mononucleotide. *J Mol Cell Cardiol*. 1983;15:537-42.
24. Kotegawa M, Sugiyama M and Haramaki N. Protective effects of riboflavin and its derivatives against ischemic reperfused damage of rat heart. *Biochem Mol Biol Int*. 1994;34:685-91.
25. Kashuro VA, Glushkov SI, Novikova TM and Aksenov VV. [Cardioprotective effect of cytoflavine on a model of doxorubicin-induced cardiomyopathy]. *Eksp Klin Farmakol*. 2010;73:15-7.
26. Jia L, Wang Y, Wang Y, Ma Y, Shen J, Fu Z, Wu Y, Su S, Zhang Y, Cai Z, Wang J and Xiang M. Heme Oxygenase-1 in Macrophages Drives Septic Cardiac Dysfunction via Suppressing Lysosomal Degradation of Inducible Nitric Oxide Synthase. *Circ Res*. 2018;122:1532-1544.
27. De Bellis A, De Angelis G, Fabris E, Cannata A, Merlo M and Sinagra G. Gender-related differences in heart failure: beyond the "one-size-fits-all" paradigm. *Heart Fail Rev*. 2019.
28. Fermin DR, Barac A, Lee S, Polster SP, Hannenhalli S, Bergemann TL, Grindle S, Dyke DB, Pagani F, Miller LW, Tan S, Dos Remedios C, Cappola TP, Margulies KB and Hall JL. Sex and age dimorphism of myocardial gene expression in nonischemic human heart failure. *Circ Cardiovasc Genet*. 2008;1:117-25.

29. Boheler KR, Volkova M, Morrell C, Garg R, Zhu Y, Margulies K, Seymour AM and Lakatta EG. Sex- and age-dependent human transcriptome variability: implications for chronic heart failure. *Proc Natl Acad Sci U S A*. 2003;100:2754-9.
30. Kararigas G, Bito V, Tinel H, Becher E, Baczko I, Knosalla C, Albrecht-Kupper B, Sipido KR and Regitz-Zagrosek V. Transcriptome characterization of estrogen-treated human myocardium identifies myosin regulatory light chain interacting protein as a sex-specific element influencing contractile function. *J Am Coll Cardiol*. 2012;59:410-7.
31. Heidecker B, Lamirault G, Kasper EK, Wittstein IS, Champion HC, Breton E, Russell SD, Hall J, Kittleson MM, Baughman KL and Hare JM. The gene expression profile of patients with new-onset heart failure reveals important gender-specific differences. *Eur Heart J*. 2010;31:1188-96.
32. Ventura-Clapier R, Moulin M, Piquereau J, Lemaire C, Mericskay M, Veksler V and Garnier A. Mitochondria: a central target for sex differences in pathologies. *Clin Sci (Lond)*. 2017;131:803-822.
33. Lagranha CJ, Deschamps A, Aponte A, Steenbergen C and Murphy E. Sex differences in the phosphorylation of mitochondrial proteins result in reduced production of reactive oxygen species and cardioprotection in females. *Circ Res*. 2010;106:1681-91.
34. Bordallo J, Secades L, Bordallo C, Cantabrana B and Sanchez M. Influence of gender and sex hormones on 5alpha-dihydrotestosterone elicited effect in isolated left atria of rats: Role of beta-adrenoceptors and ornithine decarboxylase activity. *Eur J Pharmacol*. 2009;604:103-10.
35. Bennett BJ, de Aguiar Vallim TQ, Wang Z, Shih DM, Meng Y, Gregory J, Allayee H, Lee R, Graham M, Crooke R, Edwards PA, Hazen SL and Lusis AJ. Trimethylamine-N-oxide, a metabolite associated with atherosclerosis, exhibits complex genetic and dietary regulation. *Cell Metab*. 2013;17:49-60.
36. Li XS, Obeid S, Klingenberg R, Gencer B, Mach F, Raber L, Windecker S, Rodondi N, Nanchen D, Muller O, Miranda MX, Matter CM, Wu Y, Li L, Wang Z, Alamri HS, Gogonea V, Chung YM, Tang WH, Hazen SL and Luscher TF. Gut microbiota-dependent trimethylamine N-oxide in acute coronary syndromes: a prognostic marker for incident cardiovascular events beyond traditional risk factors. *Eur Heart J*. 2017;38:814-824.
37. Tang WHW, Li DY and Hazen SL. Dietary metabolism, the gut microbiome, and heart failure. *Nat Rev Cardiol*. 2019;16:137-154.
38. Wang Z, Klipfell E, Bennett BJ, Koeth R, Levison BS, Dugar B, Feldstein AE, Britt EB, Fu X, Chung YM, Wu Y, Schauer P, Smith JD, Allayee H, Tang WH, DiDonato JA, Lusis AJ and Hazen SL. Gut flora metabolism of phosphatidylcholine promotes cardiovascular disease. *Nature*. 2011;472:57-63.
39. Zhu W, Gregory JC, Org E, Buffa JA, Gupta N, Wang Z, Li L, Fu X, Wu Y, Mehrabian M, Sartor RB, McIntyre TM, Silverstein RL, Tang WHW, DiDonato JA, Brown JM, Lusis AJ and Hazen SL. Gut Microbial Metabolite TMAO Enhances Platelet Hyperreactivity and Thrombosis Risk. *Cell*. 2016;165:111-124.
40. Bienvenu LA, Morgan J, Reichelt ME, Delbridge LMD and Young MJ. Chronic in vivo nitric oxide deficiency impairs cardiac functional recovery after ischemia in female (but not male) mice. *J Mol Cell Cardiol*. 2017;112:8-15.
41. Casin KM, Fallica J, Mackowski N, Veenema RJ, Chan A, St Paul A, Zhu G, Bedja D, Biswal S and Kohr MJ. S-Nitrosoglutathione Reductase Is Essential for Protecting the Female Heart From Ischemia-Reperfusion Injury. *Circ Res*. 2018;123:1232-1243.
42. Shao Q, Fallica J, Casin KM, Murphy E, Steenbergen C and Kohr MJ. Characterization of the sex-dependent myocardial S-nitrosothiol proteome. *Am J Physiol Heart Circ Physiol*. 2016;310:H505-15.
43. Lee WI, Xu Y, Fung SM and Fung HL. eNOS-dependent vascular interaction between nitric oxide and calcitonin gene-related peptide in mice: gender selectivity and effects on blood aggregation. *Regul Pept*. 2003;110:115-22.
44. Nuedling S, Karas RH, Mendelsohn ME, Katzenellenbogen JA, Katzenellenbogen BS, Meyer R, Vetter H and Grohe C. Activation of estrogen receptor beta is a prerequisite for estrogen-dependent upregulation of nitric oxide synthases in neonatal rat cardiac myocytes. *FEBS Lett*. 2001;502:103-8.

45. Sweet ME, Cocciolo A, Slavov D, Jones KL, Sweet JR, Graw SL, Reece TB, Ambardekar AV, Bristow MR, Mestroni L and Taylor MRG. Transcriptome analysis of human heart failure reveals dysregulated cell adhesion in dilated cardiomyopathy and activated immune pathways in ischemic heart failure. *BMC Genomics*. 2018;19:812.
46. Sun H, Olson KC, Gao C, Prosdocimo DA, Zhou M, Wang Z, Jeyaraj D, Youn JY, Ren S, Liu Y, Rau CD, Shah S, Ilkayeva O, Gui WJ, William NS, Wynn RM, Newgard CB, Cai H, Xiao X, Chuang DT, Schulze PC, Lynch C, Jain MK and Wang Y. Catabolic Defect of Branched-Chain Amino Acids Promotes Heart Failure. *Circulation*. 2016;133:2038-49.
47. Colak D, Alaiya AA, Kaya N, Muiya NP, AlHarazi O, Shinwari Z, Andres E and Dzimiri N. Integrated Left Ventricular Global Transcriptome and Proteome Profiling in Human End-Stage Dilated Cardiomyopathy. *PLoS One*. 2016;11:e0162669.
48. Gerszten RE and Wang TJ. The search for new cardiovascular biomarkers. *Nature*. 2008;451:949-52.